# Temporal shifts in 24 notifiable infectious diseases in China before and during the COVID-19 pandemic

Kangguo Li [1,5], Jia Rui[1,5], Wentao Song[1,5], Li Luo[2,5], Yunkang Zhao[1], Huimin Qu[1], Hong Liu [1], Hongjie Wei [1], Ruixin Zhang [1], Buasiyamu Abudunaibi[1], Yao Wang[1], Zecheng Zhou[1], Tianxin Xiang[3,4] ✉ & Tianmu Chen [1] ✉

The coronavirus disease 2019 (COVID-19) pandemic, along with the implementation of public health and social measures (PHSMs), have markedly reshaped infectious disease transmission dynamics. We analysed the impact of PHSMs on 24 notifiable infectious diseases (NIDs) in the Chinese mainland, using time series models to forecast transmission trends without PHSMs or pandemic. Our findings revealed distinct seasonal patterns in NID incidence, with respiratory diseases showing the greatest response to PHSMs, while bloodborne and sexually transmitted diseases responded more moderately. 8 NIDs were identified as susceptible to PHSMs, including hand, foot, and mouth disease, dengue fever, rubella, scarlet fever, pertussis, mumps, malaria, and Japanese encephalitis. The termination of PHSMs did not cause NIDs resurgence immediately, except for pertussis, which experienced its highest peak in December 2023 since January 2008. Our findings highlight the varied impact of PHSMs on different NIDs and the importance of sustainable, long-term strategies, like vaccine development.

During the coronavirus disease 2019 (COVID-19) pandemic, the emergence of various severe acute respiratory syndrome coronavirus 2 (SARS-CoV-2) variants has reshaped the transmission dynamics of other infectious diseases. Notably, influenza activity in the United States experienced a significant 98% reduction before May 2020[1]. This decline is largely attributed to the widespread implementation of public health and social measures (PHSMs) to combat COVID-19. For instance, the widespread use of face masks and enforcement of physical distancing have effectively reduced the transmission of respiratory infectious diseases (RIDs) by limiting the spread of respiratory droplets[2,3]. Additionally, travel restrictions have successfully controlled the spread of bloodborne and sexually transmitted diseases

(BSTDs), and zoonotic infectious diseases (ZIDs)[4-6]. Furthermore, improved hand hygiene has indirectly contributed to a 31% decrease in the transmission of intestinal infectious diseases (IIDs)[7,8].

Although many published studies[9-13] have analyzed the impact of PHSMs on notifiable infectious diseases (NIDs), there remains a significant gap in our understanding of this relationship. Most of these studies[9,10] have focused primarily on the impact of PHSMs in their early stages and have tended to overlook the potential impact of PHSMs of varying durations. Additionally, most published studies have predominantly concentrated on the impact of PHSMs on common RIDs and IIDs, with limited quantitative analysis on BSTDs and ZIDs[11,12]. After the Chinese government ended its "dynamic zero-COVID" policy in

[1]State Key Laboratory of Vaccines for Infectious Diseases, Xiang An Biomedicine Laboratory, State Key Laboratory of Molecular Vaccinology and Molecular Diagnostics, National Innovation Platform for Industry-Education Integration in Vaccine Research, School of Public Health, Xiamen University, Xiamen, China. [2]Health Care Departmen, Women and Children's Hospital, School of Medicine, Xiamen University, Xiamen, China. [3]Jiangxi Medical Center for Critical Public Health Events, The First Affiliated Hospital, Jiangxi Medical College, Nanchang University, Nanchang, China. [4]Jiangxi Hospital of China–Japan Friendship Hospital, Nanchang, China. [5]These authors contributed equally: Kangguo Li, Jia Rui, Wentao Song, Li Luo. ✉e-mail: ndyfy02258@ncu.edu.cn; chentianmu@xmu.edu.cn

October 2023[14], an epidemic of Omicron BA.2 variant emerged. However, there has been limited research analyzing the patterns of other infectious diseases throughout this period.

In this study, based on the NIDs data provided by the Chinese Center for Disease Control and Prevention (CDC) from 2008 to 2019, we employed multiple time series models, such as the neural network model, Bayesian structural time series model, prophet model, exponential smoothing (ETS) model, seasonal autoregressive integrated moving average (SARIMA) model, and hybrid model that combine SARIMA, ETS, STL (seasonal and trend decomposition using loess), and neural network components, to analyze the transmission trends of 24 NIDs. The objective was to forecast the transmission trends of 24 NIDs without PHSMs and SARS-CoV-2 transmission from 2020 to 2023, followed by a comparison with real-world data to analyze the impact of NIDs during different periods. Additionally, we conducted a cluster analysis to identify NIDs susceptible to PHSMs, and a cross-correlation analysis to decipher the relationship between PHSMs stringency index and the impact on NIDs.

## Results

From January 2008 to December 2023, 105,647,377 cases of 24 NIDs were reported in Chinese mainland. IIDs were the most prevalent (45.24%), followed by BSTDs (31.10%) and RIDs (22.45%). The least reported were ZIDs (1.21%). HFMD, hepatitis B, infectious diarrhea, and tuberculosis were the most common diseases, accounting for 76.87% of cases (Fig. 1A).

### Long-term NID trends before the COVID-19 pandemic

During the pre-epidemic period, the seasonality of IIDs mainly increases during summer and fall, but this pattern is not universal for

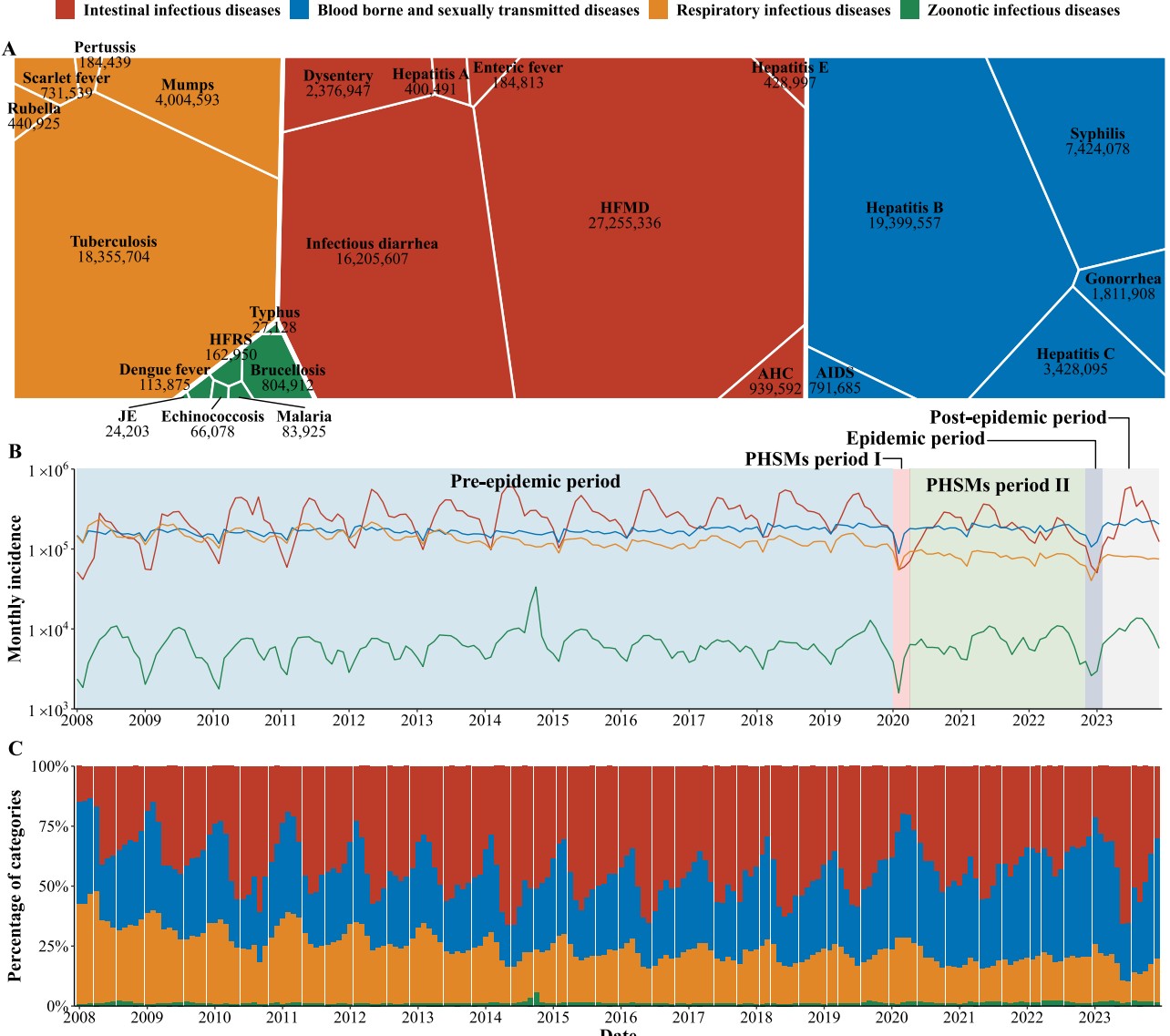

**Fig. 1 | Temporal trends and cumulative incidence of four categories of notifiable infectious diseases (NIDs) in China from January 2008 to December 2023.** **A** Cumulative incidence of 24 NIDs categorized by their respective modes of transmission, over the period from January 2008 to December 2023. The size and color of each block represent the cumulative cases and disease group, respectively. AIDS (acquired immune deficiency syndrome), not including human immunodeficiency virus infections. Dysentery includes bacterial dysentery and ameba dysentery. Enteric fever is also known as typhoid fever and paratyphoid fever. HFRS hemorrhagic fever with renal syndrome; JE Japanese encephalitis; HFMD hand, foot and mouth disease; AHC acute hemorrhagic conjunctivitis. **B** Epidemic curves for the four categories of NIDs were segmented into 5 distinct periods: pre-epidemic period (January 2008 to December 2019), PHSMs period I (January 2020 to March 2020), PHSMs period II (April 2020 to October 2022), epidemic period (November 2022 to January 2023), and post-epidemic period (February 2023 to December 2023). **C** Percentage of monthly incidences for the four groups of NIDs.

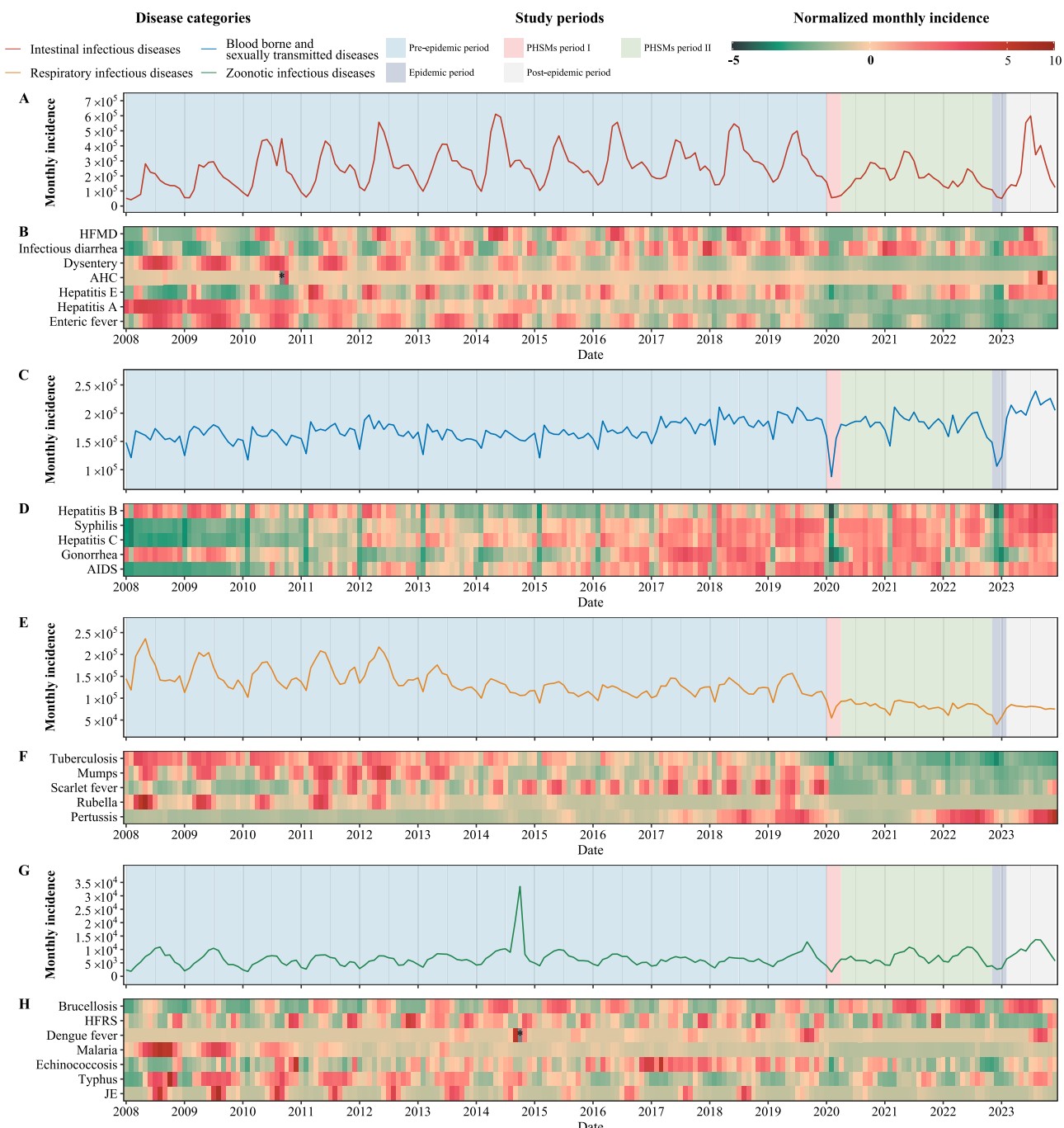

**Fig. 2 | Temporal variation in the monthly incidence of notifiable infectious diseases (NIDs) in China from January 2008 to December 2023. A**, **B** Intestinal infectious diseases. **C**, **D** Bloodborne and sexually transmitted diseases. **E**, **F** Respiratory infectious diseases. **G**, **H** Zoonotic infectious diseases. **A**, **C**, **E**, **G** Epidemic curves in the study period were segmented into 5 distinct periods: pre-epidemic period (January 2008 to December 2019), PHSMs period I (January 2020 to March 2020), PHSMs period II (April 2020 to October 2022), epidemic period (November 2022 to January 2023), and post-epidemic period (February 2023 to December 2023). **B**, **D**, **F**, **H** The normalized monthly incidence for each NIDs, with color intensity indicating the magnitude of the normalized monthly incidence. Normalization is calculated as the difference between the monthly NID incidence and the monthly average incidence, divided by the standard deviation of the incidence. Values outside the −5 to 10 range are denoted by a black box, with those exceeding 10 marked by *.

all IIDs (Fig. 2A, B). Specific diseases such as hepatitis E, dysentery, and enteric fever each exhibited distinct seasonal fluctuations. Hepatitis E peaked from January to May (Supplementary Fig. 5), while enteric fever and dysentery peaked from May to November (Supplementary Fig. 31, Supplementary Fig. 27). HFMD was notable for its biannual peaks and the remarkable alternating pattern it exhibited across odd and even years. Due to the greater incidence of HFMD than other IIDs, this biennial alteration heavily influenced the aggregate trend observed in

IIDs (Fig. 2B). There was a notable disparity in HFMD incidence between the northern and southern regions of China, with southern provinces reporting considerably more cases than their northern counterparts (Supplementary Fig. 1). Additionally, hepatitis A demonstrated more pronounced seasonality before 2012, which subsequently diminished, accompanied by a steady decline in monthly incidence after 2012 (Supplementary Fig. 30). A particularly severe outbreak from September to October 2010 led to 273,924 reported

acute hemorrhagic conjunctivitis (AHC) infections across China (Supplementary Fig. 28). All regions except for Tibet experienced a sore in AHC, and peaks in the remaining 30 provinces lasted 1–2 months (Supplementary Fig. 4). Guangxi and Guangdong provinces had most cases, which are 79,977 and 69,839 cases, respectively, during this period. For detailed information, see Supplementary Data 1.

BSTDs demonstrated a recurrent annual trough in February, yet the epidemiological data did not reveal distinct seasonal peaks. The monthly incidence of these diseases ranged between 10,000 and 20,000 cases from 2008 to 2016. A notable inflection point occurred in 2017 when more than 2.1 million cases were reported, and a sustained increase in incidence was documented (Figs. 1B and 2D). Specifically, the annual reported cases of syphilis and hepatitis C increased for 11 consecutive years during pre-epidemic periods, averaging yearly increases of 7.22% and 7.46%, respectively. Furthermore, the incidence of AIDS increased rapidly, increasing from 12,409 cases in 2008 to 72,630 cases in 2019, with an average annual increase of 17.43% (Supplementary Fig. 36). Regional disparities in AIDS reporting were evident, initially prominent in Henan and Guangxi provinces, and a shift in the following years as Sichuan province surpassed Guangxi after 2014, peaking at 17,869 cases in 2019 (Supplementary Fig. 12). Hepatitis B, the most reported BSTD, primarily concentrated in Henan and Guangdong provinces, experienced brief declines reported cases between 2013 and 2016. However, by 2019, the incidence approached 2011 levels 2019 (1,247,092 vs. 1,252,236) (Supplementary Fig. 8). Moreover, fluctuating pattern observed in BSTDs was mainly driven by hepatitis B, which had peak seasons in March each year since 2012 and continued until 2020. Notably, hepatitis B, syphilis, hepatitis C, gonorrhea, and AIDS collectively experienced a cyclic trough every February from 2010 to 2019, which was concentrated in 31 provinces (Fig. 2D, Supplementary Figs. 8–12).

Conversely, RIDs maintained a maximal seasonal variation, with monthly incidences varying from 100,000 to 250,000 cases (Fig. 2E,F). Rubella was a significant component of RIDs before 2013. However, its incidence and seasonality have gradually decreased. A rubella outbreak from March to June 2019 resulted in 25,736 reported cases nationwide (Supplementary Fig. 16). All provinces, excluding Zhejiang, Tibet, Tianjin, and Qinghai, saw an increase in cases, with Chongqing (4334 cases) and Hunan (3733 cases) reporting the most cases. Tuberculosis also exhibited a noteworthy seasonal pattern, with the fewest cases reported annually in January or February (Supplementary Fig. 13). However, at the onset date level, the lowest incidence was typically recorded in November and December each year. Mumps and scarlet fever both showed two peaks each year, from April to July and November to January, with notable regional differences. Mumps was mainly concentrated in Guangdong, with a sharp increase in Henan and Hunan from 2017 to 2019 (Supplementary Fig. 14). Scarlet fever, primarily in Shandong, grew steadily from 28,507 cases in 2008 to 83,028 cases in 2019 (Supplementary Fig. 15). Notably, pertussis cases surged from under 5000 annually during 2008-2014 to 30,727 in 2019, particularly in Shandong and Guangdong (Supplementary Fig. 17). For more detailed information, please refer to Supplementary Table 1.

The seasonal patterns of ZIDs from 2008 to 2020 were similar to those of IIDs but with less pronounced peaks. Four out of the seven ZIDs (excluding dengue fever and echinococcosis) displayed clear seasonality (Fig. 2G, H). Typhus was prevalent year-round, peaking in autumn before 2014, mainly in Yunan province (Supplementary Fig. 23). In malaria, the gradual success of eradication efforts has been reflected in a decreasing incidence trend, particularly after 2011 (Supplementary Fig. 21). The peak seasons of ZIDs shifted, with brucellosis, hemorrhagic fever with renal syndrome (HFRS), and Japanese encephalitis (JE) becoming predominant. These diseases had similar numbers of reported cases (Fig. 1A) but peak seasons spanning March to July, October to January, and June to September, respectively (Supplementary Figs. 18, 19 and 24). Which led to a more homogenized seasonality curve, lacking the

traditional peaks associated with ZIDs. Notably, an outlier of ZIDs in 2014 primarily attributed to a dengue fever outbreak in Guangdong province (Figs. 1B, 2G, Supplementary Fig. 20). Reported cases in September and October 2014 surged to 14,759 and 28,796 respectively, a significant increase from the 1289 and 1473 cases in the same months of 2013 (Supplementary Fig. 44, Supplementary Data 1). For more detailed information, please refer to Supplementary Table 1.

## Short-term NID trends during the COVID-19 pandemic

Before 2020, BSTDs and RIDs showed an increase in the proportion of cases during winter and spring, accounting for more than 50% of NIDs. This fell to less than 50% in summer and autumn due to the prevalence of IIDs. However, reported IID cases significantly decreased during the PHSMs period I (from January 2020 to March 2020) and PHSMs period II (from April 2020 to October 2022), while reported BSTD/RID cases quickly resurged after an initial decline, exceeding 50% of reported cases during the PHSMs periods (Fig. 1C).

During PHSMs period I, the government implemented numerous PHSMs, and all NIDs entered a low-prevalence phase (Fig. 2). Distinguishing whether the observed incidence decreases were due to the diseases' inherent characteristics or the impact of PHSMs was challenging. To address this, separate time series models were developed, with the optimal model selected based on the greatest composite standardized index. This index integrated root mean square error (RMSE), mean absolute percentage error (MAPE), and symmetric mean absolute percentage error (SMAPE). In the test dataset, neural network models were optimal for enteric fever and HFRS (Fig. 3G, S). ETS and SARIMA models excelled for three NIDs (Fig. 3C, F, P) and eight NIDs (Fig. 3B, J, K, L, O, Q, T, W), respectively. The hybrid model also applied to eight NIDs, including HFMD, AHC, hepatitis E, syphilis, mumps, brucellosis, malaria, and JE (Fig. 3A, D, E, I, N, R, U, X). Additionally, the Bayesian structural model was effective for diseases including hepatitis B, tuberculosis, and echinococcosis (Fig. 3H, M, V). However, the Prophet model, despite its capabilities, was not selected as the optimal model for any of the 24 NIDs (Fig. 3).

By comparing the forecasted result with the real-world data during PHSMs period I, except for hepatitis A, which increased by 522 cases (14.70%) (Fig. 4F), the other 23 NIDs saw case reductions ranging from 21.01% to 70.46%. HFMD, dengue fever, rubella, and scarlet fever experienced the most significant decreases (Fig. 4A, T, P, O), with drops of 82,279 (−70.46%), 245 (−70.19%), 3466 (−67.07%) and 11,460 (−60.84%) cases, respectively (Fig. 5). RIDs were the most affected, with wide heterogeneity in the adjusted incidence relative ratio (IRR), ranging from 0.16 to 0.74. Among them, the impact on tuberculosis was lowest, with a median adjusted IRR of 0.74 (interquartile range [IQR]: 0.69–0.81) (Fig. 4M, Supplementary Table 2). In contrast, the impact on the BSTDs was relatively stable, with the median adjusted IRRs ranging from 0.50 to 0.74 (Fig. 5F, Supplementary Table 2).

Since April 2020, with the adoption of the "dynamic zero-COVID" policy and the relaxation of intercity travel restrictions by the Chinese government, the epidemiological patterns of diseases have entered a new phase, PHSMs period II[14]. This period is characterized by distinct trends in disease incidence, which can be broadly categorized into 4 types. The most common type includes diseases such as infectious diarrhea (Fig. 4B), dysentery (Fig. 4C), AHC (Fig. 4D), hepatitis A (Fig. 4F), enteric fever (Fig. 4G), syphilis (Fig. 4I), hepatitis C (Fig. 4J), AIDS (Fig. 4L), tuberculosis (Fig. 4M), mumps (Fig. 4N), scarlet fever (Fig. 4O) and JE (Fig. 4X). These diseases demonstrated seasonal or irregular patterns with an overall decreasing trend in cases. The second type is commonly observed among ZIDs and BSTDs which either maintained their seasonal or irregular patterns with a minor overall change (<10%) or experienced an increase in reported cases. This type includes hepatitis B (Fig. 4H), gonorrhea (Fig. 4K), brucellosis (Fig. 4R), HFRS (Fig. 4S), echinococcosis (Fig. 4V) and typhus (Fig. 4W). The third type includes NIDs like HFMD (Fig. 4A), hepatitis E (Fig. 4E), pertussis

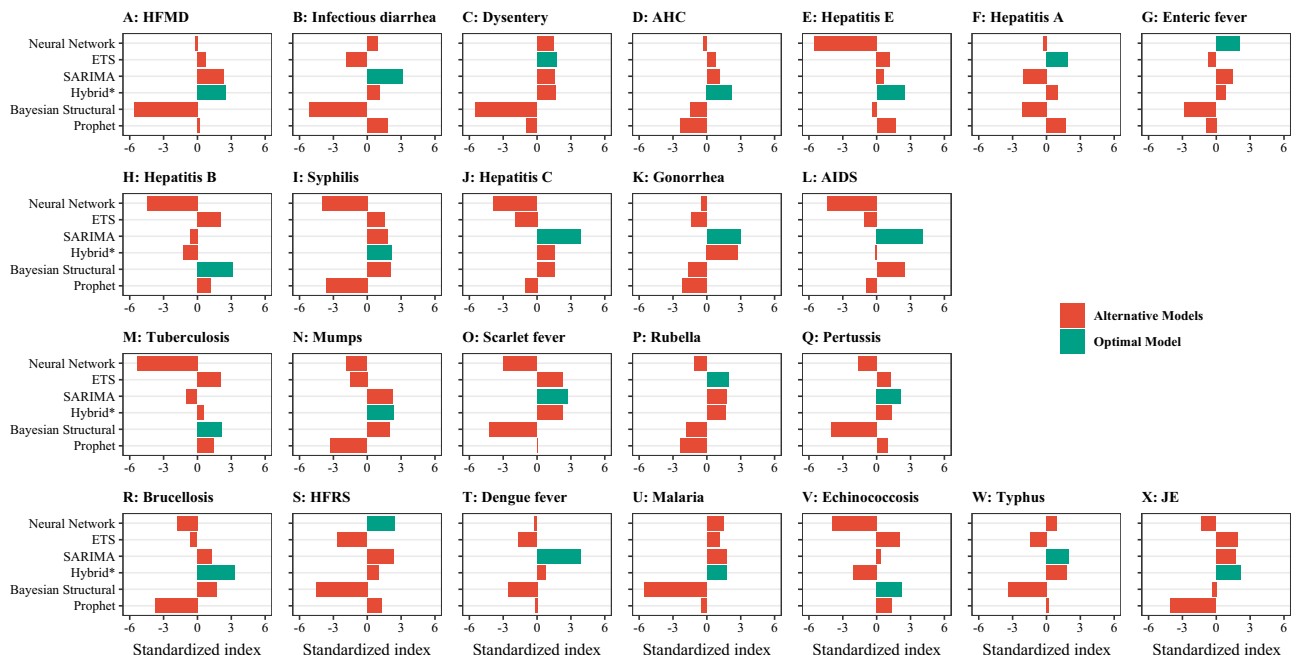

**Fig. 3 | Comparative performance analysis of various time series models on the monthly incidence data for 24 notifiable infectious diseases (NIDs) in the test dataset. A–G** Intestinal infectious diseases. **H–L** Bloodborne and sexually transmitted diseases. **M–Q** Respiratory infectious diseases. **R–X** Zoonotic infectious diseases. The comparative standardized index was developed by integrating the root mean square error (RMSE), mean absolute percentage error (MAPE), and symmetric mean absolute percentage error (SMAPE), where a higher value signifies the optimal model for a specific disease. ETS exponential smoothing; SARIMA

seasonal autoregressive integrated moving average; Hybrid: combined SARIMA, ETS, STL (seasonal and trend decomposition using loess), and neural network models; AIDS (acquired immune deficiency syndrome) not including human immunodeficiency virus infections. Dysentery includes bacterial dysentery and ameba dysentery. Enteric fever is also known as typhoid fever and paratyphoid fever. HFRS hemorrhagic fever with renal syndrome, JE Japanese encephalitis, HFMD hand, foot and mouth disease, AHC acute hemorrhagic conjunctivitis.

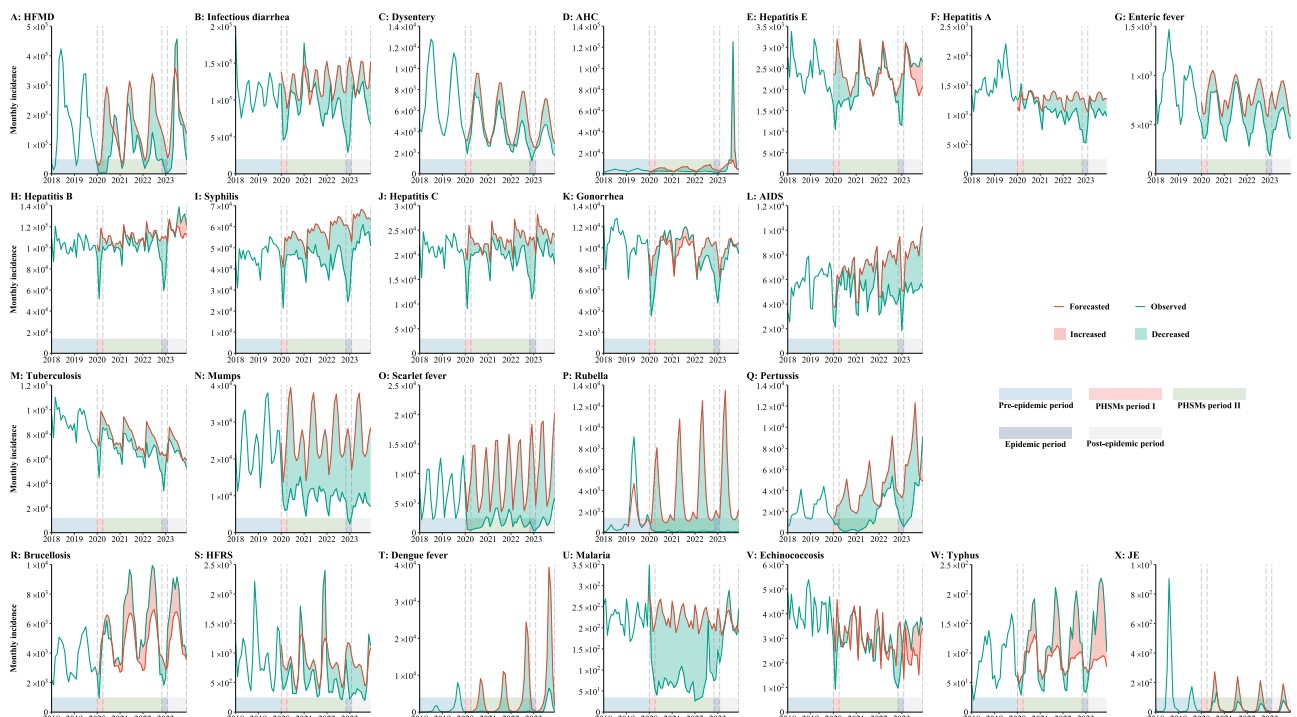

**Fig. 4 | Forecasted and actual incidence of 24 notifiable infectious diseases (NIDs) in China. A–G** Intestinal infectious diseases. **H–L** Bloodborne and sexually transmitted diseases. **M–Q** Respiratory infectious diseases. **R–X** Zoonotic infectious diseases. AIDS (acquired immune deficiency syndrome), not including human immunodeficiency virus infections. Dysentery includes bacterial dysentery and ameba dysentery. Enteric fever is also known as typhoid fever and paratyphoid

fever. HFRS hemorrhagic fever with renal syndrome, JE Japanese encephalitis, HFMD hand, foot and mouth disease, AHC acute hemorrhagic conjunctivitis. Each panel is divided into five periods: pre-epidemic period (January 2008 to December 2019), PHSMs period I (January 2020 to March 2020), PHSMs period II (April 2020 to October 2022), epidemic period (November 2022 to January 2023), and post-epidemic period (February 2023 to December 2023).

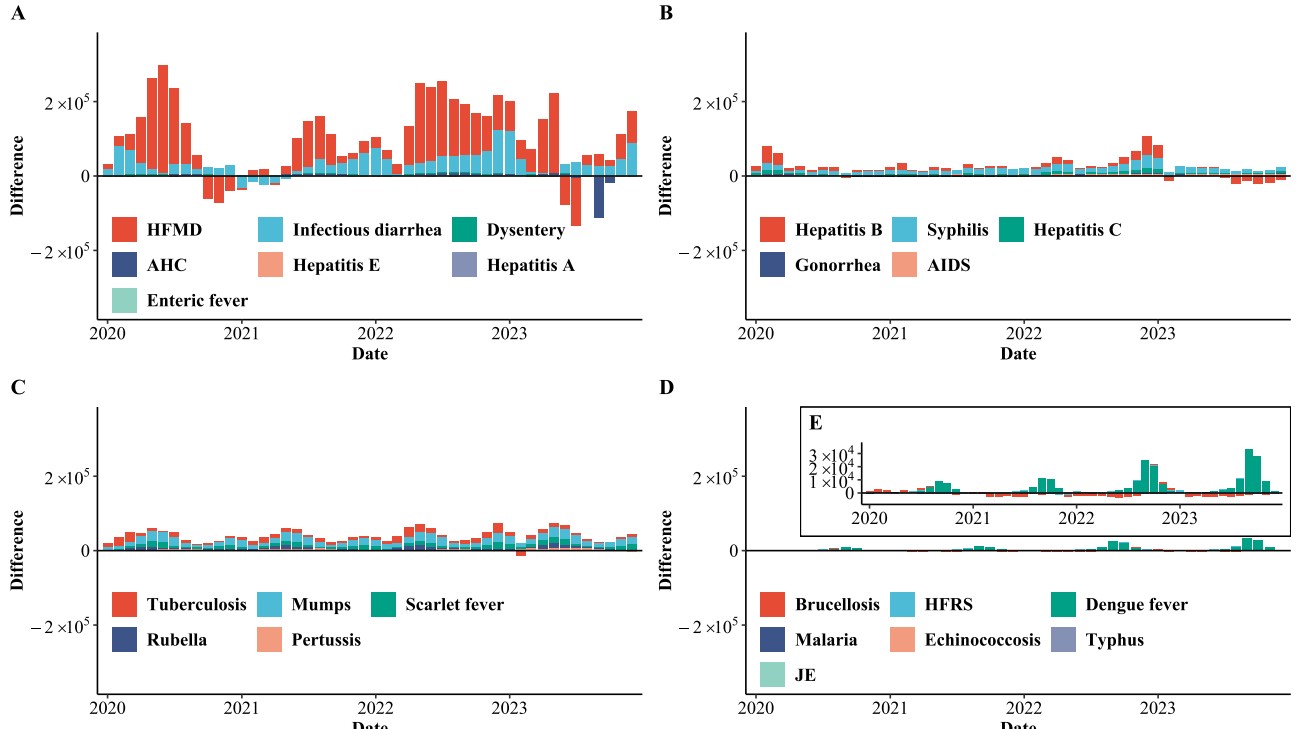

**Fig. 5 | Differences between the forecasted and actual incidence of 24 notifiable infectious diseases (NIDs) in China. A** Intestinal infectious diseases. **B** Bloodborne and sexually transmitted diseases. **C** Respiratory infectious diseases. **D**, **E** Zoonotic infectious diseases. AIDS (acquired immune deficiency syndrome), not including human immunodeficiency virus infections. Dysentery includes bacterial dysentery and ameba dysentery. Enteric fever is also known as typhoid fever and paratyphoid fever. HFRS hemorrhagic fever with renal syndrome, JE Japanese encephalitis, HFMD hand, foot and mouth disease, AHC acute hemorrhagic conjunctivitis.

(Fig. 4Q), and malaria (Fig. 4U), which initially showed a decrease in prevalence but subsequently exhibited a gradual return to their normal trends. Distinct from other NIDs, rubella, and dengue fever presented a unique trend during PHSMs period II, with reported cases nearing zero (Fig. 4P, T). The rubella dataset used for testing in the first stage and retraining in the second stage excluded the 2019 data due to the significant impact of the 2019 rubella outbreak on the models (Supplementary Fig. 50). However, even with this exclusion, the difference between observed data and forecasted outcomes revealed a significant impact on rubella during this period (Fig. 4P).

In December 2022, the Chinese government stopped the "dynamic zero-COVID" policy[14]. Similar to trends during PHSMs period I, 23 NIDs except for JE saw a sharp initial decline in incidence (Fig. 2). Even brucellosis, which had consistently high growth during PHSMs period II, saw its adjusted *IRR* drop to 0.58 in December 2022 (Fig. 6H). The incidence of HFMD during the epidemic period was greater than during PHSMs period I (80,864 vs. 34,487) (Fig. 4A), while for other IIDs, it was lower (Fig. 4B-G, Supplementary Table 2). Among RIDs, pertussis was a special disease with a higher incidence than in PHSMs period I (4,336 vs. 2,753) (Fig. 6Q, Supplementary Table 2). However, following the SARS-CoV-2 variant epidemic in the Chinese mainland, some NIDs reverted to pre-epidemic patterns. HFMD (Fig. 5A), AHC (Fig. 5A), and hepatitis B (Fig. 5B) showed signs of resurgence, notably with a substantial AHC outbreak in September 2023, resulting in 125,264 cases. This outbreak, second only to the September 2010 outbreak, predominantly affected the southern provinces, with Guangdong being the epicenter. (Supplementary Fig. 4). Additionally, pertussis reached its highest incidence registered since 2008, with 9,126 cases from October to December 2023, also concentrated in Guangdong (Supplementary Fig. 17).

In summary, the incidence of NIDs notably declined during the PHSMs and epidemic periods. Among BSTDs, AIDS experienced the most significant reduction, with a median adjusted *IRR* of 0.73 (IQR:

0.59–0.80, *p* < 0.01) during PHSMs and epidemic periods, which further decreased to 0.64 (IQR: 0.59–0.66, *p* = 0.001) in the post-epidemic period (Fig. 6C). RIDs also experienced substantial declines, with tuberculosis showing the smallest reduction (0.88, IQR: 0.84–0.92, *p* < 0.001), while other RIDs displayed notable and sustained decreases even in the post-epidemic period (Fig. 6E). Rubella and dengue fever cases plummeted, with the former showing adjusted *IRR*s as low as 0.05 (IQR: 0.02–0.08, *p* < 0.001) during previous periods and 0.03 (IQR: 0.02–0.06, *p* = 0.001) in the epidemic period; the adjusted *IRR* for dengue fever was 0.01 (IQR: 0–0.03, *p* < 0.001) during previous periods and 0.10 (IQR: 0.03–0.16, *p* = 0.001) thereafter. Conversely, the adjusted *IRR* of ZIDs such as brucellosis, malaria, echinococcosis, typhus, and JE reported increased (Fig. 6G-H). All IIDs had adjusted *IRR* lower than 1, except for hepatitis E, which initially had a median adjusted *IRR* of 0.91 (IQR: 0.77–0.99, *p* < 0.001) during PHSMs and epidemic periods, but in the post-epidemic period, the median adjusted *IRR* increased to 1.17 (IQR: 1.04–1.28, *p* = 0.05), indicating a rebound in the incidence (Fig. 6A).

**Relationship between PHSMs strength and *IRR***

Clustering analysis of monthly incidence during pre-epidemic period categorized hepatitis B, tuberculosis, infectious diarrhea, and HFMD into a high-incidence group, a stark contrast to the 20 NIDs classified as low-incidence group (Figs. 7A and 1A). Refined clustering based on adjusted *IRR*s identified 4 discrete clusters, with dengue fever and rubella as individual clusters due to their distinct epidemiological patterns (Fig. 7B), while scarlet fever, pertussis, HFMD, mumps, malaria, and JE formed a cluster suggestive of shared PHSMs susceptibility. The other 16 NIDs appeared to be less affected by PHSMs (Fig. 7C). Clusters 1, 3, and 4, predominantly comprising RIDs and ZIDs, were significantly responsive to PHSMs, with no representation from the BSTD category, mirroring the general trend (Fig. 7B).

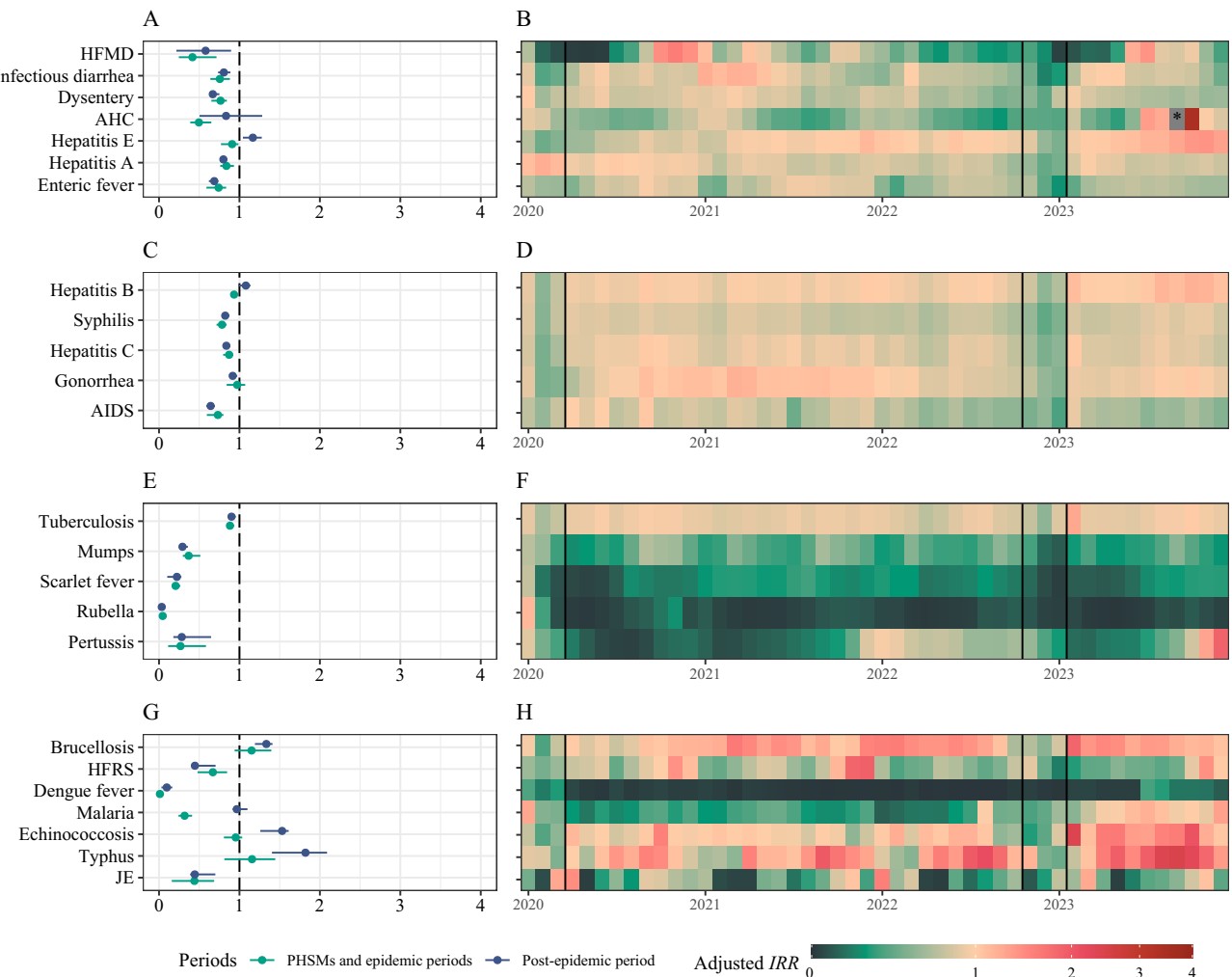

**Fig. 6 | Adjusted incidence relative ratios (IRR) of 24 notifiable infectious diseases (NIDs) in China from January 2020 to December 2023. A, B** Distribution and changes in relative ratios for intestinal infectious diseases. **C, D** Distribution and changes in relative ratios for bloodborne and sexually transmitted diseases. **E, F** Distribution and changes in relative ratios for respiratory infectious diseases. **G, H** Distribution and changes in relative ratios for zoonotic infectious diseases.

**A, C, E, G** The distribution of the adjusted relative ratio of 24 NIDs during different periods. The point and line depict the median value and interquartile range (IQR), respectively. The PHSMs and epidemic periods range from January 2020 to January 2023 ($n = 37$), and the post-epidemic period ranges from February 2023 to December 2023 ($n = 11$).

Focusing on HFMD, a notable cumulative decrease of 2,652,880 cases (48.86%) was observed during PHSMs periods, indicating a high incidence coupled with substantial PHSMs susceptibility (Supplementary Table 2). Seven other NIDs displayed both low incidence and high PHSMs sensitivity, including dengue fever, rubella, scarlet fever, pertussis, mumps, malaria, and JE. Cross-correlation analysis indicated that the correlation between PHSMs stringency and the incidence differences for dengue fever (Fig. 7D), scarlet fever (Fig. 7F), HFMD (Fig. 7H), mumps (Fig. 7I), malaria (Fig. 7J), and JE (Fig. 7K) occurred in the same month. Nevertheless, only malaria showed a moderate correlation coefficient of 0.42 without lag time, suggesting a direct temporal relationship with PHSMs stringency (Fig. 7J). The correlation between PHSM stringency and incidence for rubella peaked, with a correlation coefficient of 0.14, after a 3-month lag, decreasing to 0.10 without delay (Fig. 7E). Notably, pertussis exhibited a maximum correlation coefficient of 0.08 at a one-month lag, decreasing further to 0.06 with no lag, both of which are indicative of a minimal association (Fig. 7G).

## Discussion

In this study, we conducted a modeling study on 24 NIDs, and analyzed transmission characteristics across 5 periods: pre-pandemic period,

PHSMs period I, PHSMs period II, epidemic period, and post-epidemic period. Furthermore, cluster and cross-correlation analysis were used to evaluate the impact of PHSMs on NIDs dynamics. Our findings revealed distinct seasonal patterns in different NID, varying across the studied periods. 24 NIDs except for hepatitis A experienced significant reductions during PHSMs period I and the epidemic period. During the PHSMs period II, only hepatitis B, gonorrhea and certain ZIDs like brucellosis, HFRS, echinococcosis, and typhus showed limited declines, other NIDs experienced more than 10% decreases.

Furthermore, a comparative analysis of forecasted and observed data revealed 8 NIDs susceptible to PHSMs, including dengue fever, rubella, scarlet fever, pertussis, HFMD, mumps, malaria, and JE (Fig. 7D–K). However, only HFMD, mumps, malaria, and JE exhibited moderate or weak correlation, while the other 4 NIDs showed no association with the PHSMs stringency index with an absolute correlation coefficient less than 0.2. Dengue fever and rubella (Fig. 7D, E) were consistently classified as distinct categories, whether they were observed in PHSMs period II where they are categorized as NIDs with reported cases nearing zero, or in cluster analysis. This is mainly because both the actual incidence and adjusted *IRR* are close to zero, regardless of how the PHSMs stringency index varied (Fig. 6E, G).

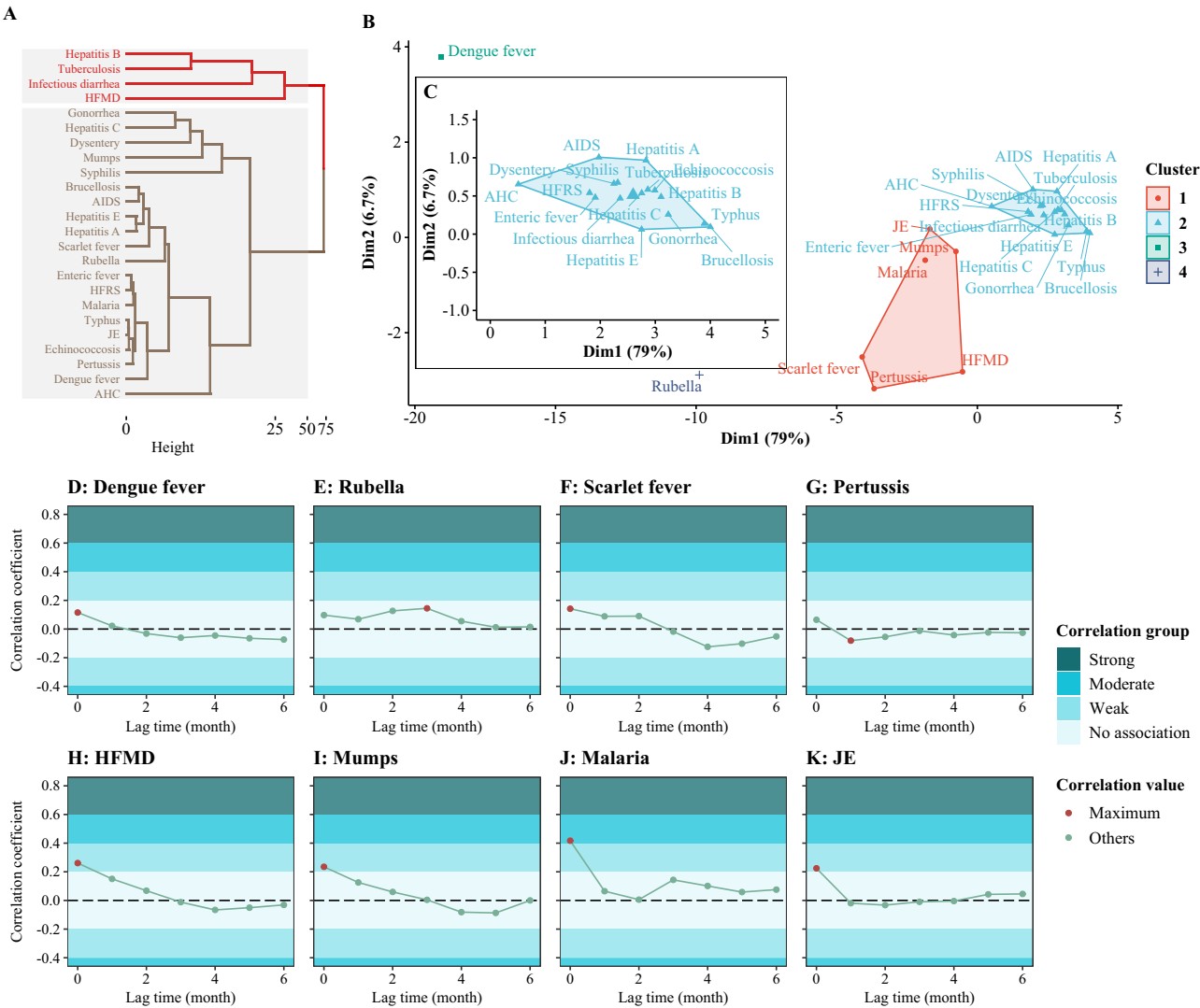

**Fig. 7 | Cluster and cross-correlation analysis of notifiable infectious diseases (NIDs) susceptible to PHSMs.** **A** Cluster tree of the monthly incidence of 24 NIDs during pre-epidemic period (January 2008 to December 2019). The red part represents the high-incidence cluster, and the brown represents the low-incidence cluster. **B, C** Cluster scatterplot of the logarithmically adjusted incidence relative ratios (IRR) of 24 NIDs during PHSMs periods (January 2020 to October 2023).

**D–K** Cross-correlation analysis between the reduction in incidence and the PHSMs stringency index. AIDS (acquired immune deficiency syndrome), not including human immunodeficiency virus infections. Dysentery includes bacterial dysentery and ameba dysentery. Enteric fever is also known as typhoid fever and paratyphoid fever. HFRS hemorrhagic fever with renal syndrome, JE Japanese encephalitis, HFMD hand, foot and mouth disease, AHC acute hemorrhagic conjunctivitis.

Furthermore, scarlet fever and pertussis showed low correlation coefficients due to the pronounced seasonality in the disparity between observed and forecasted incidence. Specifically, they displayed significant variations during May to June and July to August in the PHSMs period, respectively, contrasting with minor differences in other months (Fig. 5C, Supplementary Fig. 49). Such seasonality undermines the correlation between the PHSMs stringency index and the incidence difference. To elucidate the impact of rubella, we forecasted the incidence from 2019 to 2023 based on the ETS, SARIMA, and hybrid models, which performed well on the test dataset with a similar composite standardized index (1.92 vs. 1.73 vs. 1.63) (Fig. 3R). Findings from these models consistently demonstrated a significant decline in the prevalence of rubella, as evidenced by the median of the adjusted *IRR* falling below 0.2 during the PHSMs and epidemic periods (Supplementary Fig. 49).

Among the NIDs that were susceptible to PHSMs, HFMD was the only disease with a high incidence and susceptible to PHSMs, showing substantial declines throughout most periods, except for unexpected seasonal peaks from October to November 2020 and June to July 2023

(Fig. 5A). These irregular patterns in HFMD suggest altered epidemiological trends not only during PHSMs periods but also in the post-epidemic period. The anomalies observed need to be validated with data from 2024 to determine whether these anomalies are amplified minor peaks or shifts in the occurrence of major peaks typically expected in even-numbered years from June to July. Notably, only malaria exhibited a moderate correlation coefficient of 0.42 without a lag time (Fig. 7J). The decrease in malaria cases was mainly attributed to strict travel restrictions. Malaria is a mosquito-borne disease that has been eliminated in China and is primarily spread by international travelers who are infected in other countries[9].

Cross-correlation analysis revealed that the highest relative correlation coefficients for JE, HFMD, mumps, and malaria occurred within the same month, suggesting that these diseases were immediately impacted following the implementation of PHSMs. This impact can be attributed to the immediate effectiveness of PHSMs in controlling the spread of these diseases, which are primarily transmitted through direct contact and respiratory droplets. The rapid response of these diseases to PHSMs may also be related to their relatively short

incubation periods, which allows changes in transmission dynamics following the implementation of PHSMs to be quickly reflected. However, JE, mumps, scarlet fever, and rubella demonstrated susceptibility to PHSMs, and no resurgence exceeding the forecasted or epidemic trends during the pre-epidemic period. This decline cannot be solely attributed to PHSMs or the influence of the National Notifiable Diseases Surveillance System (NNDSS). The increased vaccination coverage of meningococcal polysaccharide vaccine (MPV) and measles, mumps, and rubella (MMR) vaccines likely explains the trends observed in JE, mumps and rubella cases[15,16]. Moreover, the incidence of scarlet fever not only has been gradually increasing in China (Fig. 4O), but has also increased in multiple European countries between September and November 2022[17]. This emphasizes the need for continued surveillance and research on its transmission dynamics.

The observed seasonal variations in NIDs observed during the pre-epidemic period are primarily due to the interplay between transmission models and behavioral patterns[18]. RIDs such as mumps show a higher prevalence during the winter[19,20], a trend attributed to increased frequency of indoor gatherings in enclosed spaces without physical distancing, which facilitates virus transmission[21]. Additionally, the low humidity and cooler temperature in winter increase vulnerability to RIDs of the upper respiratory tract[22]. Third, seasonal variations may also impact immune responses, potentially increasing susceptibility to infections at certain times of the year[23,24]. NIDs experience substantial decrease during PHSMs period I (Fig. 1B), which can be attributed to a confluence of factors. The advent of the COVID-19 pandemic significantly heightened public awareness and vigilance towards infectious diseases, prompting the widespread adoption of handwashing and mask-wearing[8], which may have directly reduced exposure to pathogens. Although residents have maintained good hygiene habits since April 2020, the decline in some NIDs such as HFMD and infectious diarrhea is starting to shrink (Fig. 5). This observation may be due to the relaxation of PHSMs, leading to increased social interactions, population migration and potential for NID transmission[9,11,25]. Similar to PHSMs period I, a reduction in NIDs also observed during the epidemic period, which may attributed to the decreased mobility of Omicron BA.2 variant infections[25], coupled with an increased propensity to wear masks. Additionally, compliance with PHSMs extended even to susceptible individuals, even in the absence of government enforcement. This collective adherence significantly contributed to the observed decrease in the transmission of respiratory viruses[26], underscoring the effectiveness of PHSMs in the containment of infectious diseases.

Acknowledging the impacts on NID incidence during the COVID-19 pandemic is essential. The observed changes in NIDs cannot be solely attributed to actual shifts in incidence but are also likely affected by factors leading to increased underreporting. These factors include disruptions in the NNDSS and shifts in healthcare-seeking behaviors. Specifically, the high-intensity and redirection work of grassroots health personnel to COVID-19 prevention and control tasks may have resulted in missed reports of other diseases. Additionally, the severity of the pandemic altered healthcare-seeking behavior, with some individuals reducing their frequency of medical consultations. A study in China reported a 26.17% decrease in outpatient visits during February–June 2020[26].

The impact of PHSMs varied across diseases with different modes of transmission. RIDs, notably HFMD and mumps, displayed a pronounced susceptibility to PHSMs[27]. Conversely, the impact of PHSMs on BSTDs was more limited, consistent with previous research findings[9,10,26]. For most RIDs, the implementation of PHSMs significantly curtailed their transmission (Fig. 5E), reaffirming previous studies on the efficacy of PHSMs in controlling these diseases[9,26]. This reduction can be attributed to the primary transmission route of these diseases, i.e., respiratory droplets, which can be effectively managed by measures such as mask-wearing, physical distancing, and improved

ventilation. In contrast, BSTDs, which are primarily transmitted through direct contact with infected bodily fluids, may not be as effectively mitigated by these measures[26]. It is important to note, however, that while the median adjusted *IRR*s for AIDS, syphilis, hepatitis C, and hepatitis B were generally less than 1 during PHSMs and epidemic periods (Fig. 6C), indicating a reduced impact, this does not imply that these diseases were entirely unaffected. Interestingly, the gonorrhea incidence increased during 2021 (Fig. 4K), potentially due to its short incubation period[28], which resulted in quick responses to PHSMs. In contrast, for diseases such as AIDS, syphilis, hepatitis C, and hepatitis B, which typically exhibit longer incubation periods (over a month)[29], a substantial delay between symptom onset and reporting may occur.

Travel restrictions significantly affect the transmission of ZIDs like dengue fever and malaria, which are mosquito-borne and sensitive to imported cases[25,30]. In the winter, the inhospitable condition for mosquitos help suppress these diseases across most of China[31]. During the PHSMs periods, stringent international travel restrictions, combined with the synchronization between the isolation period and the incubation period for imported cases of dengue fever and malaria, frequently facilitated the detection of these cases during quarantine[6,9]. Consequently, local outbreaks of these diseases markedly decreased during PHSMs periods. However, brucellosis (Fig. 4R), another ZID, significantly increased during PHSMs period II, particularly in northern provinces such as Xinjiang, Gansu (Supplementary Fig. 18), and Inner Mongolia[32]. This rise is primarily due to inadequate animal vaccination and a meat shortage that spurred the expansion of sheep herds, practitioners, and home poultry farming, thereby increasing human-animal contact and the risk of brucellosis[32,33]. Another area requiring attention is typhus which is a disease transmitted by parasites. Since 2017, typhus incidence has shown a steadily increasing trend (Fig. 4W). This rise could be attributed to factors such as changes in human behavior that increase exposure to vectors, urbanization, or climate change[34]. Further investigation into the specific causes of this increase is necessary to develop effective control strategies.

The significant resurgence of AHC and pertussis in the post-pandemic period is noteworthy. Previously exhibiting a relatively low incidence, both diseases experienced unexpected large-scale rebounds after the epidemic. AHC, primarily caused by enteroviruses, instigated an outbreak in southern China in September 2023. Notably, AHC outbreaks also occurred in Pakistan and India due to Coxsackievirus A24 in September and July, respectively[35,36]. This unexpected escalation of AHC and pertussis may be attributed to several factors, including relaxed PHSMs, increased susceptibility due to COVID-19, or even potential viral mutations. Pertussis reached its highest incidence since 2008 in December 2023 in China and is experiencing a global resurgence[37]. Alongside the aforementioned factors, decreased population immunity due to under-vaccination may also contribute to the resurgence of pertussis[37].

Our study has several limitations. First, we focused on 24 NIDs instead of all infectious diseases, leaving a particular gap in the study of respiratory diseases. Since influenza was excluded due to its reliance on sentinel surveillance[38], this does not undermine the overall conclusion of the study. RIDs continue to exhibit high incidence and are notably susceptible to PHSMs. The selection of 24 high-incidence NIDs for this study is indicative of the broader trends observed across all NIDs. Additionally, our analysis relies on monthly NIDs reports, which are updated frequently but potentially introduce reporting delays, particularly for diseases such as tuberculosis. To address this, we incorporated data from slower-updating sources (Supplementary Figs. 25–48), finding no significant seasonality variations in onset and reporting dates for the other 23 NIDs. However, discrepancies in reported cases were observed for some diseases (e.g., hepatitis B, hepatitis C, tuberculosis) (Supplementary Figs. 8, 10, 13), potentially due to data collection errors, statistical differences, or other factors.

These discrepancies have not been previously reported, and further attention is warranted. Further investigation and official clarification are crucial for reconciling these discrepancies and ensuring data accuracy. However, our analyses were mainly based on reports, with disease onset data analysis primarily supplemented through geographical distribution analysis, mitigating the impact of these discrepancies on our main conclusions. We also developed a website application to facilitate reader analysis based on customized data (https://kanggle.shinyapps.io/auto-tsmodel/). Third, the inherent limitations of time-series models must be acknowledged. These limitations include the inability to capture sudden, unexpected events and the direct impact of intervention measures on disease transmission. These models also rely on assumptions about data being stationary and following certain trends. Therefore, when comparing our results with those from other studies, it is essential to consider these limitations with due diligence.

Our research revealed that the implementation of PHSMs in response to various SARS-CoV-2 variants can significantly impact the transmission dynamics of most NIDs. Intriguingly, despite the relaxation of all PHSMs by the Chinese government, there was no immediate significant resurgence in NIDs during the epidemic period. This funding suggests that widespread self-isolation practices after the Omicron BA.2 epidemic had a temporary restraining effect on the transmission of other infectious diseases. However, this period of restricted transmission was followed by increased reported cases of diseases such as HFMD, AHC and pertussis. Other studies have also reported a 74.8–140.1% increase in influenza infections during the 2022–2023 season[39]. This phenomenon, referred to as the "immune gap," highlights the "broad-spectrum" effectiveness of PHSMs. While they mitigated SARS-CoV-2 transmission, PHSMs also reduced exposure to various pathogens. This reduced exposure may have led to insufficient immune system stimulation, potentially weakening population immunity compared to pre-epidemic levels. Our findings emphasize that while PHSMs offer an effective short-term solution for controlling the spread of infectious diseases, their long-term use has the unintended consequence of potentially decreasing population immunity, which could create conditions for future large-scale outbreaks. A more sustainable, long-term approach prioritizes the development and widespread implementation of effective vaccines, similar to the successful model of the MPV and MMR vaccine[15,16].

# Methods

## Disease selection criteria
The disease selection criteria were determined based on data collected by China's NNDSS, which includes information from 31 provinces, excluding the Hong Kong Special Administrative Region (SAR), Macau SAR, and Taiwan province. The NNDSS was created in 2004 and has been significantly improved to monitor multiple NIDs[9]. As of December 2023, data from 41 NIDs were included, but only 30 NIDs had more than 20,000 cases reported during the study period (from January 2008 to December 2023). COVID-19 and 5 other NIDs were excluded due to data deficiencies. Specifically, influenza was excluded from the analysis because its detection depends on specialized sentinel surveillance systems[38]. Influenza A(H1N1) was virtually eliminated, and it was removed from monthly reports after November 2013 (Supplementary Data 1). "Other hepatitis" encompassing cases clinically diagnosed but not confirmed as hepatitis types A, B, C, D, or E, were also excluded from our analysis. Because the incidence of "other hepatitis" has been significantly influenced by the evolution of laboratory and hospital testing capabilities[40], technological advancements have led to a progressive decline of 'other hepatitis' cases (Supplementary Data 1). Moreover, diseases such as schistosomiasis and measles, which are nearing elimination in China and have been recently characterized by a relatively low prevalence[41], were also not considered in the analysis.

This study adopts a modeling methodology to examine the patterns of 24 NIDs including HFMD, infectious diarrhea, dysentery, AHC, hepatitis E, hepatitis A, enteric fever, hepatitis B, syphilis, hepatitis C, gonorrhea, AIDS, tuberculosis, mumps, scarlet fever, rubella, pertussis, brucellosis, HFRS, dengue fever, malaria, echinococcosis, typhus and JE. These diseases were categorized into 4 categories based on their primary modes of transmission: IIDs, BSTDs, RIDs, and ZIDs. Specifically, IIDs include HFMD, infectious diarrhea, dysentery, AHC, hepatitis E, hepatitis A and enteric fever. BSTDs include hepatitis B, syphilis, hepatitis C, gonorrhea and AIDS. RIDs include tuberculosis, mumps, scarlet fever, rubella and pertussis. Finally, ZIDs encompass brucellosis, HFRS, dengue fever, malaria, echinococcosis, typhus and JE (Fig. 1A).

## Data collection
National data for this study were systematically collected from the monthly NIDs reports published by the National Health Commission of China. These reports, aggregating data from the NNDSS based on reported date, have been available since January 2004. However, due to the instability of the NNDSS data sources in its early years, our analysis did not include reports prior to January 2008. The study period spans from January 2008 to December 2023. Notably, for specific NIDs, such as HFMD, AHC, infectious diarrhea, mumps, rubella, echinococcosis and typhus, data were collected by the NNDSS from January 2008 to February 2009 but were not reflected in the monthly NIDs Reports. For these diseases, we relied on data provided by the Chinese Public Health Science Data Center (CPHSDC), maintained by the Chinese CDC, which also aggregates data from the NNDSS based on onset date and includes early NID data. However, it is notable that the monthly NID reports are frequently updated the following month to reflect the reported date of cases, while data provided by the CPHSDC is typically updated at a slower pace, sometimes taking ~3–4 years. Considering the relatively short incubation period of these 7 NIDs and no significant incidence difference from 2010 to 2021 (Supplementary Figs. 1, 2, 4, 14, 16, 22 and 23), it is evident that the influence of these data on the research findings is minimal.

Additionally, the CPHSDC provided provincial-level NIDs data until December 2020. For provincial data beyond this date, we extracted information from the monthly NIDs reports published by provincial CDCs or health commissions. Due to discrepancies in data availability and presentation across provinces, only 11 provinces publicly provided complete data tables or figures from January 2021 onwards: Anhui, Chongqing, Gansu, Guangdong, Henan, Jiangsu, Shandong, Shanghai, Sichuan, Xinjiang, and Zhejiang. Analysis of the remaining 20 provinces did not yield complete data records due to various factors, including the absence of detailed NID tables ($n = 17$) or restricted public access ($n = 3$).

The study period was divided into distinct periods to align with the different phases of the SARS-CoV-2 epidemic in China. The period spanning from January 2008 to December 2019 is designated the pre-epidemic period, reflecting the epidemiological landscape before the emergence of SARS-CoV-2. The advent of the epidemic and the subsequent PHSMs response were segmented into several phases. The period from January 2020 to March 2020 marks PHSMs period I, reflecting the early response to the outbreak[9]. This period is followed by the period from April 2020 to October 2022, termed PHSMs period II, characterized by sustained measures aimed at controlling the spread of the virus. The distinction between PHSMs period I and PHSMs period II is primarily based on the cessation of lockdowns in Wuhan, marking a significant shift in control strategies[9]. The transition to the epidemic period, spanning November 2022 to January 2023, corresponds to the Chinese government's shift away from the "dynamic COVID-zero" strategy[14]. The final phase, from February 2023 to December 2023, is recognized as the post-epidemic period,

demarcated by a notable decrease in the positive rate of COVID-19 tests, indicating a reduction in viral transmission[42].

Considering the varying intensities of PHSMs across these different periods, we utilized the stringency index to measure the national intensity of PHSMs. This composite index is based on 13 policy response indicators, including school closures, workplace closures, travel bans, testing policies, contact tracing, face coverings, and vaccine policies. The index is normalized to a scale ranging from 0 to 100, providing a robust measure of PHSM intensity over time[43]. Detailed information about the sources of NID data and the website links for monthly NID reports can be found in Supplementary Data 1. To guarantee data accuracy and reliability in this analysis, all NID datasets underwent a randomized, double-blind verification process by multiple authors.

## Model building

A single time series model alone is insufficient for capturing the epidemic patterns of all 24 diseases due to the diverse epidemiological characteristics and temporal distributions of different infectious diseases. Therefore, the ensemble forecasts include the neural network model, Bayesian structural time series model, Prophet model, ETS model, SARIMA model, and hybrid model (combine SARIMA, ETS, STL and neural network components). Each model has advantages depending on the specific epidemic characteristics of different diseases. The neural network model excels in capturing nonlinear trends and complex relationships. The Prophet model automatically handles long-term trends, seasonality, and holiday effects. Bayesian structural time series models address uncertainty and randomness; the ETS model is suitable for smoothing data and short-term forecasting; and the SARIMA model considers trends, seasonality, and autoregressive terms simultaneously. By combining the weighted averages of the neural network, STL, ETS, and SARIMA models, a hybrid model can better capture the epidemic trends of different infectious diseases.

**Neural network model.** We utilized a feed-forward neural network with lagged inputs and a single hidden layer, containing half as many neurons as the input layer. Multiple networks were trained with distinct initial random weights, and their forecasts were averaged. The network was calibrated for single-step predictions, while multistep projections were derived recursively[44].

**ETS model.** The three parameters of the ETS model were automatically determined using a log-likelihood optimization criterion, guided by the corrected Akaike information criterion (AICc)[44].

**SARIMA model.** The SARIMA model parameters $(p, d, q) \times (P, D, Q)s$ were systematically selected using a stepwise algorithm informed by the AICc, facilitated by the "auto.arima" function of the "forecast" package in R (version 4.3.2, R Core Team, Vienna, Austria)[44].

**Hybrid model.** Our hybrid approach synthesizes the predictive power of the SARIMA, ETS, STL, and neural network models. By assigning weights to the forecast of each base model according to its out-of-sample error rate and normalizing them to sum to unity, we recalibrate these weights annually to better align with the evolving disease patterns[45].

**Bayesian structural time series model.** These models were executed using the "bsts" package in R (version 4.3.2, R Core Team, Vienna, Austria), incorporating structural components for trend, seasonality, and regression effects. Priors were selected via the empirical Bayes method, and we conducted 500 MCMC simulations to ensure convergence[46].

**Prophet model.** Prophet employs an additive model to fit nonlinear trends with components for yearly, weekly, and daily seasonality, as well as holidays. The Prophet model, available in R, excels with data exhibiting strong seasonal patterns and copes well with missing data, trend shifts, and atypical values[47].

To estimate the baseline epidemic trend of the 24 NIDs from January 2020 to December 2023 (without PHSMs and SARS-CoV-2 transmission), this study used a two-phase modeling approach using historical NID incidence data. The monthly NID incidence from January 2008 to December 2017 constituted the training dataset for the baseline models. The subsequent period, from January 2018 to December 2019, was utilized as the test dataset to evaluate the predictive performance of these models. Laplace smoothing was applied in two-phase modeling, involving the addition of 0.1 cases to the incidence data for each month, to address zero-reported cases in certain months. This adjustment not only resolved the zero-incidence issue but also improved model stability by ensuring no null input data points. Upon model construction, the forecasted outcome will decrease by 0.1.

To evaluate the performance of time series models on the test datasets, we employed three evaluation metrics: RMSE, MAPE and SMAPE. RMSE is valuable for its ability to highlight significant errors, indicating the seriousness of substantial deviations in predictions. However, the RMSE lacks directional sensitivity and may exaggerate the impact of large outliers. The scale-invariance and measurement of errors in percentages of MAPE aid in interpretability, although its accuracy diminishes when the actual values approach zero, leading to a potential distortion of errors. The SMAPE addresses this issue by equally penalizing overpredictions and underpredictions but is still sensitive to outliers and may face challenges in situations where both actual values and forecasts are zero. In this study, we aggregated these metrics into a single composite index of equal weight. Additionally, these indicators, each with unique range sensitivities, underwent a transformation process to standardize their values for comparative analysis using z-normalization. The comprehensive standardized index was delineated by following formula:

$$Z_{m,d} = -\sum_i \frac{x_{m,i,d} - \mu_{i,d}}{\sigma_{i,d}} \qquad (1)$$

Here, $Z_{m,d}$ represents the standardized index for model $m$ in disease $d$, and $x_{m,i,d}$ indicates the value of index $i$ (RMSE, MAPE or SAMPE) for model $m$ in disease $d$. $\mu_{i,d}$ and $\sigma_{i,d}$ denote the mean and standard deviation of index $i$ in disease $d$, respectively. This normalization process ensures that the performance of each model across different diseases and indices is comparable. The model with the highest standardized index for a given disease across 6 models was considered the optimal model for that disease within this analysis. Additionally, to enhance the reliability of our analysis across various models for different NIDs, we employed cross-validation techniques. This involved dividing the training data into subsets for model fitting and evaluation. Training datasets spanned 6 to 10 years, with the test datasets being established 2 years after the training dataset (Supplementary Table 3).

## Statistical analysis

After establishing the optimal model, we retrained the model using data from 2008 to 2019. The model was then applied to forecast the incidence of various infectious diseases since January 2020. By comparing the forecasted results with the real-world incidence, we calculated the *IRR*s of PHSMs for different NIDs. The formula for calculating the *IRR* is the ratio of the incidence during PHSMs to the incidence without PHSMs:

$$IRR = \frac{I_a}{I_m} \qquad (2)$$

In this study, the time series model employed occasionally produced negative values, mainly when the incidence was low. All negative values in the analysis were adjusted to 0 to address this issue. To

further refine our approach and mitigate the issue of zero incidence values, we applied Laplace smoothing to adjust the *IRR* calculations:

$$IRR_{adj} = \frac{I_a + 0.1}{I_m + 0.1} \qquad (3)$$

The adjusted *IRR* determines the impact of PHSMs on the risk of disease incidence. An adjusted *IRR* less than 1 indicates that PHSMs can reduce the risk, while an adjusted *IRR* greater than 1 indicates that PHSMs can increase the risk. The significance of the difference between the adjusted *IRR* and 1 was assessed using two-sample Wilcoxon tests with a significance level of 0.05. Furthermore, to categorize NIDs into distinct groups based on their incidence in the pre-epidemic period and adjusted *IRR* in PHSMs periods, cluster analysis was applied. This analysis used hierarchical k-means clustering based on Euclidean distance and Ward.D2 method, which was first utilized to split the hierarchical tree into k clusters. Subsequently, the centroids of each cluster were calculated and served as the initial cluster center for the k-means clustering algorithm based on Euclidean distance[48]. Based on incidence data from 2008 to 2019, NIDs were divided into high and low incidence categories through this cluster analysis. Additionally, through the clustering analysis of logarithmically adjusted *IRR*s, diseases were roughly classified into two categories: NIDs that are less susceptible to PHSMs, and those that are more susceptible to PHSMs.

To comprehensively understand the impact of PHSMs on the incidence of NIDs, cross-correlation analysis was conducted to assess the relation between the stringency of PHSMs and the logarithms of adjusted *IRR*s. Considering the variations in the incubation periods of different infectious diseases and the potential lag effects of PHSMs, we utilized a cross-correlation analysis to analyze the combined effects of the monthly stringency index, both without and with a lag of 1 to 6 months, on adjusted *IRR*[49]. The aim of this comprehensive approach was to dissect the temporal associations and causative relationships between PHSM stringency levels and NID incidence fluctuations, employing a systematic categorization of correlation coefficients to delineate the magnitude of associations: unrelated (below 0.2), weak (0.2 to 0.4), moderate (0.4 to 0.6), and strong (above 0.6)[50].

Ethical approval was not required for the data used in this study. All the data of NIDs are publicly available on the websites of Health Commission, CDC, and CPHSDC. The PHSMs stringency index is also publicly available in Our world in data (https://ourworldindata.org/covid-stringency-index). All the statistical analyses were conducted using R (version 4.3.2; R Core Team, Vienna, Austria).

### Reporting summary
Further information on research design is available in the Nature Portfolio Reporting Summary linked to this article.

## Data availability
The monthly incidence data of 24 NIDs used in this study is accessible through the GitHub repository (https://github.com/xmusphlkg/code_PHSM), and Supplementary Data 1. National notifiable infectious disease incidence data in China also available at figshare (https://doi.org/10.6084/m9.figshare.24589608). Source data are provided in this paper.

## Code availability
The R code used for statistical analysis and figure generation is also available at the GitHub repository (https://github.com/xmusphlkg/code_PHSM) and Zenodo[51].

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

## Acknowledgements

The authors extend their gratitude to all CDC staff members who contributed their expertise and support to this project. Special thanks are due to Kun Su from the Chongqing Provincial CDC for providing valuable insights and guidance. This study was partly supported by the Self-supporting Program of Guangzhou Laboratory (SRPG22-007) which granted for T.M.C., the National Key Research and Development Program of China (2021YFC2301604) which granted for T.M.C., the Fundamental Research Funds for the Central Universities (20720230001) which granted for TMC, the Fujian Science and Technology Development Funds (2023Y0004) which granted for T.M.C., and the Provincial Key Research and Development Program of Jiangxi, China (20232BBG70020) which granted for T.X.X. It is also important to recognize the contributions of GitHub Student Developer Pack, whose support has been invaluable in completing this analysis.

## Author contributions

Conceptualization: T.M.C., T.X.X., K.G.L., J.R., L.L. Investigation: K.G.L., J.R., W.T.S., L.L., Y.K.Z., H.M.Q., H.L., H.J.W., R.X.Z., B.A., Y.W., Z.C.Z. Methodology: K.G.L., L.L., J.R., T.M.C., H.J.W. Software: K.G.L., Y.K.Z., H.J.W., Z.C.Z., H.L. Validation: W.T.S., K.G.L., L.L., T.X.X. Writing—original draft: K.G.L., J.R., L.L. Writing—review & editing: T.M.C., W.T.S., R.X.Z., B.A., Y.W., H.M.Q., T.X.X. All authors read and approved the final manuscript.

## Competing interests

The authors declare no competing interests.
