## [Peer Review File · Nature Communications]

Temporal shifts in 24 notifiable infectious diseases in China before and during the COVID-19 pandemicREVIEWER COMMENTS

Reviewer #1 (Remarks to the Author):

The present study conducted an analysis on the impact of Public Health and Social Measures (PHSMs) on 24 Notifiable Infectious Diseases (NIDs) in China, employing multiple time series models to forecast transmission trends under hypothetical scenarios without PHSMs or pandemics. Generally, respiratory infectious diseases exhibited a significant response to PHSMs, while bloodborne and sexually transmitted diseases demonstrated more moderate effects. Furthermore, PHSMs such as travel restrictions played an important role in mitigating the transmission of zoonotic diseases like dengue fever and malaria. This study is intriguing and holds relevance for a broader readership. However, the COVID-19 pandemic has significantly disrupted various systems and healthcare-seeking behaviors, potentially altering the patterns of NIDs. It would be beneficial to further investigate this aspect or acknowledge its impact in the discussion. Additionally, it is important to note that certain respiratory infectious diseases or zoonotic diseases exhibit distinct seasonal variations across different regions, which may not be adequately represented when combined together.

Specific comments:

Lines 134-138. This result is puzzling; seasonality is usually the most significant for respiratory infectious diseases, especially when individual infectious disease is presented separately.

Lines 155-156. "The best model was selected using the normalized composite index", It is not clear how the best model was selected based on the normalized composite index from Fig 3? The greatest value of the normalized composite index? please provide the details.

Fig. 1A: "the cumulative infections (2018.1-2023.1): The two circles, each representing 2,000,000 cases but of different sizes, are depicted here. please correct; "Level of Infectious Disease": Here the capitalization of every word is not required. Similarly, "Monthly Incidence (Normalized)" in Fig 2A.

The complete name of the abbreviation presented in each figure should be included in the corresponding figure legends, e.g., providing full disease names for abbreviations.

Reviewer #2 (Remarks to the Author):

This study described the incidence patterns of 24 notifiable infectious diseases (NIDs) during 2008 and early 2023 based on public available NIDs data reported in mainland China. By using multiple time series models, they also compared the observed rates and predicted rates to impact of implementation of public health and social measures (PHSMs) and epidemic of COVID-19 in the incidence of different NIDs. Some major comments here:

1. Given the NIDs in China are required to be reported within 24 hours and the Reports are published monthly, it would be better to include a longer duration (e.g. 6 more months) of incidence data of those 24 NIDs after March 2023 in the analyses. This would allow them to check whether the seasonality patterns changed after the epidemic, particularly if wanted to claim the statement in the last paragraph of Discussion section.
2. Authors used first 10 years (2008-2017) data as training set and 2 more years (2018-2019) data as a testing set to define the model for forecasting the rate during PHSMs and endemic period. More information should be provided on the definition of optimal model and how reliable of comparing the results fitted by the different model for different NIDs? Some results were quite surprising or unexpected. Using rubella as an example, due to the vaccination, rubella has reached at very low incidence rate since 2013 but suddenly had a high pick in 2019 then back to nearly zero level since 2020 (Supplementary Figure 24). The paper didn't explain the unusual peak in 2019 and the forecasting model repeated that unusual pattern during 2020-2023 (Figure 4) by the optimal model selected (Bayesian structural model as shown in Figure 3).

3. Different scales used for the Y-axis in Figure 4 for different disease which may exaggerate the effects for those low prevalent NIDs.

Specific comments:

1. Page 4 line 65: please indicate when, where and what the data source are?
2. Page 4 line 79: 'The last reported' should be 'The least reported'?
3. Page 4 line 80 and other relevant places in the context: please keep consistent for using the term, either as 'zoonotic infectious disease' or 'natural focal disease'
4. Page 5 and other relevant places in context: the specific supplementary figure need to be referred clearly in the text
5. I found it's hard to follow the results presented in different orders of NIDs categories in the context as well as figures, better to keep in a consistent order
6. Page 7 in the section of 'respiratory infectious diseases': there was an abnormal peak of rubella in 2019 which should be presented/explained (Supplementary Figure 24)
7. Page 7 in the section of 'respiratory infectious diseases': what about tuberculosis which showed a regular pattern as in Supplementary Figure 22
8. Pages 7-8 lines 143 onwards: confused, not sure whether PHSMs period I and II were counted as 'before' or 'during' epidemic as data from both periods been presented in both sections?
9. Page 8 second paragraph: not sure why present certain NIDs optimal model but not others? And seems dysentery has two models been selected as optimal model: neural network (shown in both text and Figure 3) or hybrid model (shown in text line 162)
10. Page 10 lines 201-205: malaria was missed
11. Page 10 first paragraph: clarify what's the information shown in Figure 5 A C E G?
12. Page 11 line 227: Fig. 5 I should be Fig. 5 J?
13. Page 11 line 232: not sure about the statement of HFMD 'significant susceptibility to the impact of PHSMs' which perhaps in absolute term as a highly prevalent NID, but not in relative term, due to a similar pattern found as in the pre-epidemic period
14. Page 13 lines 264-267: surprised that with such low incident rate, malaria showed a surprisingly high relative correlation (Figure 6). Moreover, other three NIDs had only very weak correlation (just above 0.2) as shown in Figure 6, thus the statement sounds not very reliable
15. Page 13 lines 275-277: there's an abnormal peak in 2019 which authors didn't present however the forecast repeated the wave of that, thus the statement may not reliable
16. Page 16 line 344: not sure how author defined 'high-incidence NIDs' in the study, some NIDs had very low rate
17. Pages 16-17 lines 345-350: not sure about this discussion point, and more information needs to be provided for the NIDs reporting system in China
18. Page 17 361 onwards: don't think this study support this argument, if wanted, the author should include those two diseases in the analyses and compare the data before and after epidemic
19. Page 19 lines 398-402: how comparable of those two datasets? More information should be provided
20. Page 19 line 411: better to include data up to October 2023 in the analyses
21. Page 20 second paragraph: reference 15 is an editorial, not relevant
22. Page 23 line 482-484: more explanations are needed for the formula included; please also clarify how the optimal model has been defined/selected?

Reviewer #3 (Remarks to the Author):

In this paper, the authors conduct a modeling study using different types of time-series models to model transmission dynamics of 24 notifiable infectious diseases (NIDs), categorized into 4 groups according to the mode of transmission that were reported in mainland China from January 2008 to March 2023 – before and after the COVID-19 pandemic measures (PHSMs). They describe disease trends, identify the best forecasting model, and evaluate the impacts of the COVID-19 PHSMs on each of the 24 NIDs. Authors conclude that while PHSMs can be an effective short-term solution for inadvertently controlling the spread of non-SARS-CoV-2 NIDs, they are not effective or sustainable in the long-term, and the development and implementation of effective vaccines should be prioritized.

This is a comprehensive study on NID trends with different modes of transmission before and during the COVID-19 pandemic and the implementation of PHSMs in China. The methods chosen are described in detail, clearly, and extensively. Results and figures are well represented. The result section shows the number of disease cases, long-term trends before the COVID-19 pandemic, and describes seasonality for each of the 24 NIDs. Six different time-series models were implemented, and the best model was identified for each disease. These models were then used to forecast the incidence of NIDs assuming the absence of SARS-CoV-2 in 2020 and onward and compare predicted incidence with the real incidence data. Models were used to describe diseases most affected by PHSMs. Discussion should be developed further to make this paper much more impactful.

Major changes:

Discussion should be developed further to make this paper much more impactful.

Lines 287-298: There is a substantial decrease in the incidence during the first half of PSHM I, but then the trends start going back towards the baseline trends during PSHM II (except for respiratory viruses), which is not specifically discussed or clearly explained in this paragraph. Omicron BA.2 appeared in Dec 2021, much later than April 2020, which was the start of the PSHM II period, to explain this observation. How about discussing the stringency index here? Perhaps, although still in place, measures were more relaxed during PSHM II and contributed to this. The paragraph has no supporting references. Comparing to studies from other countries could be also beneficial.

Lines 324-338: Proposed explanations for the observed patterns in this paragraph lack references that can provide support. There should be some available in the recent literature.

Minor changes:

Lines 52, 55, and 60 and some other lines throughout the manuscript would benefit from making some grammar changes.

Line 64 – add “time-series” to clarify the type of models in the introduction.

Lines 129-131: The statement is confusing; I am guessing that you mean the inflection point in 2018 was mainly due to the upsurge of bloodborne and STD cases in 2018, while the other 3 disease groups maintained their fluctuating patterns and disease incidence levels.

Line 211 – Would it not now shift from “relaxed intercity travel restrictions” since it has shifted from “dynamic zero-COVID” policy to relaxed intercity travel restrictions in April 2020 (described on line 185)?

Line 688 – Is this a typographical error? Dysentery, not dynthesis

Figure 1A – spelling of “cumulative”

Reviewer #1 (Remarks to the Author):

The present study conducted an analysis on the impact of Public Health and Social Measures (PHSMs) on 24 Notifiable Infectious Diseases (NIDs) in China, employing multiple time series models to forecast transmission trends under hypothetical scenarios without PHSMs or pandemics. Generally, respiratory infectious diseases exhibited a significant response to PHSMs, while bloodborne and sexually transmitted diseases demonstrated more moderate effects. Furthermore, PHSMs such as travel restrictions played an important role in mitigating the transmission of zoonotic diseases like dengue fever and malaria. This study is intriguing and holds relevance for a broader readership.

Author response:

We are deeply grateful for your insightful feedback, which has been instrumental in refining this manuscript. Your thorough reviews and keen observations have enriched our analysis of the effects of PHSMs on NIDs and instigated substantial enhancements across the paper. We have diligently considered and implemented your suggestions, thereby improving the manuscript's clarity, precision, and relevance. Notable amendments inspired by your recommendations include:

1. Elaborating on the disparity between forecasted and observed incidence.
2. Incorporating provincial variations in 24 NIDs preceding and during the COVID-19 pandemic.
3. Separately analyzing each NID in the results section.
4. Detailing the calculation method of the normalized composite index.

A comprehensive point-by-point response to your feedback, including more specific changes made, is provided below:

Major changes:

However, the COVID-19 pandemic has significantly disrupted various systems and healthcare-seeking behaviors, potentially altering the patterns of NIDs. It would be beneficial to further investigate this aspect or acknowledge its impact in the discussion.

Author response:

Thank you very much for your insightful opinions and suggestions. We acknowledge the potential impact of the COVID-19 pandemic on the patterns of NIDs and its influence on healthcare-seeking behaviors. Our research now encompasses a comprehensive analysis of these dynamics, providing a detailed discussion of how these phenomena potentially skew the impact of PHSMs on NIDs in line 333-356.

We acknowledge that alterations in the reporting of NIDs cannot be solely attributed to straightforward changes in disease patterns. Instead, these modifications likely result from a complex interplay of factors that contribute to increased underreporting. A crucial factor among these is the disruptions encountered by the National Notifiable Diseases Surveillance System (NNDSS), coupled with significant shifts in the public's approach to seeking medical care. The redirection of grassroots healthcare personnel toward addressing COVID-19, prioritizing prevention, and control over routine surveillance, may have resulted in substantial gaps in reporting other communicable diseases. This situation underscores the critical need for healthcare systems to maintain flexibility and robustness in surveillance capacities, even during a pandemic.

Furthermore, the impact of the pandemic on societal norms and individual behaviors concerning health consultations cannot be overstated. Apprehensions surrounding COVID-19, combined with public health measures, have led to a significant decline in routine healthcare engagements, as evident in a 26.17% decrease in outpatient visits in China during the early

pandemic months¹. This shift indicates a broader trend in healthcare avoidance, where fear of infection or adherence to PHSMs prompted individuals to postpone or forego medical consultations for non-COVID conditions, further complicating the accurate assessment of NIDs prevalence during this period.

1. *Chen S, Zhang X, Zhou Y, Yang K, Lu X. COVID-19 protective measures prevent the spread of respiratory and intestinal infectious diseases but not sexually transmitted and bloodborne diseases. J Infect 83, e37-e39 (2021).*

Additionally, it is important to note that certain respiratory infectious diseases or zoonotic diseases exhibit distinct seasonal variations across different regions, which may not be adequately represented when combined together.

Author response:

We greatly appreciate your insightful feedback and the opportunity to further elaborate on our research result. Your point regarding the distinct seasonal variations of specific respiratory and zoonotic diseases across different regions is well-taken. In our study, we have indeed addressed this aspect by gathering data from 31 provinces across China. Given the inconsistencies in reporting data, both in terms of time and covered diseases among various provinces, we found that despite significant improvements by 2023, 20 provinces were either not publicly reporting data or were only providing rough aggregate data without detailed disease incidence. Thus, we relied on the data provided by the Chinese Public Health Science Data Center (https://www.phsciencedata.cn/share/ky_sjml.jsp), also maintained by the Chinese Center for Disease Control and Prevention (CDC). This data, aggregated from the NNDSS based on onset dates, includes early NID data.

Upon comparative analysis, we discovered that even though both sets of data are compiled and disseminated by the Chinese CDC, there are notable differences in the detailed quantities for certain diseases such as Hepatitis B, C, and Tuberculosis. However, we assert that these discrepancies do not significantly distort our study results. The data we used primarily contributes to the analysis of seasonal and spatial distribution, with minimal impact on model predictions and outcomes. We have emphasized this point in our limitations discussion section to provide a transparent account of our study's constraints and the potential influence of regional variations on our findings.

As you astutely noted, disease seasonality varies considerably across different regions. For instance, Hand, Foot, and Mouth Disease (HFMD), which has the highest number of reported cases among 24 NIDs, are predominantly concentrated in the southern provinces of Guangdong, Guangxi, and Hunan, with comparatively fewer reported cases in the northern regions. Regarding seasonality, southern provinces like Guangdong and Hainan report cases from March to December, while northern provinces such as Heilongjiang, Xinjiang, and Inner Mongolia primarily see cases from April to August each year. We have included further distinctions in the distribution of various diseases across different provinces and regions in the results section of our study. These nuances underscore the importance of region-specific disease surveillance and control measures.

Specific comments:

Lines 134-138. This result is puzzling; seasonality is usually the most significant for respiratory infectious diseases, especially when individual infectious disease is presented separately.

Author response:

We appreciate your comment on the perplexing outcome. We acknowledge your observation regarding the importance of seasonality for respiratory infectious diseases. In our study, we also recognize this significance, and we have taken seasonality into account as a critical element in our analysis and interpretation of the results. Moreover, it appears that there was a typo in the text, as it should have stated "With the gradual decline in rubella incidence, the seasonality of rubella decreased." Thank you for your valuable feedback; we have rectified this error in line 117-125.

Lines 155-156. "The best model was selected using the normalized composite index", It is not clear how the best model was selected based on the normalized composite index from Fig 3? The greatest value of the normalized composite index? please provide the details.

Author response:

Thank you for your question. We apologize for the lack of clarity in explaining how the best model was selected based on the normalized composite index. In our study, we selected the model with the greatest value of the normalized composite index as the optimal model. We have updated the manuscript to include this information in the result section (line 164-167) and provide more details in the method section (line 594-619). Thank you for bringing this to our attention.

Fig. 1A: “the cumulative infections (2018.1-2023.1): The two circles, each representing 2,000,000 cases but of different sizes, are depicted here. please correct; “Level of Infectious Disease”: Here the capitalization of every word is not required. Similarly, “Monthly Incidence (Normalized)” in Fig 2A.

Author response:

Thank you for bringing this to our attention. We apologize for the mistake in Fig. 1A. To further clarify, we have replaced the original bubble chart with a tree diagram and labeled the cumulative case numbers for each disease in the figure. Additionally, we have carefully checked the text in each figure and revised the labels to remove unnecessary capitalization of each word. Thank you for pointing this out.

The complete name of the abbreviation presented in each figure should be included in the corresponding figure legends, e.g., providing full disease names for abbreviations.

Author response:

Thank you for your feedback. We apologize for the oversight in not including the complete names of the abbreviations in the figure legends. We acknowledge the importance of providing clarity to the readers and ensuring that all abbreviations are properly explained. In response to your comment, we have revised the figure legends in the manuscript to include the full disease names for all abbreviations. Thank you for bringing this to our attention, and we appreciate your valuable input.

Reviewer #2 (Remarks to the Author):

This study described the incidence patterns of 24 notifiable infectious diseases (NIDs) during 2008 and early 2023 based on public available NIDs data reported in mainland China. By using multiple time series models, they also compared the observed rates and predicted rates to impact of implementation of public health and social measures (PHSMs) and epidemic of COVID-19 in the incidence of different NIDs. Some major comments here:

Author response:

We sincerely value the time and effort you have invested in reviewing our manuscript. Your expert feedback has been critical in highlighting areas needing improvement, refinement, and clarification. Following your recommendations, we have made significant modifications to the manuscript, including:

1. Extending the data to December 2023 and incorporating the post-epidemic period.
2. Revising the calculation method of the normalized composite index to select the reliable optimal model.
3. Adjusting the training dataset of the rubella model to mitigate the impact of the 2019 rubella outbreak.
4. Adding Fig. 5 to illustrate the impact on 24 NIDs using the same scale on the Y-axis.
5. Adding a new subsection to explain disease selection, data sources, and the normalized composite index.

A detailed point-by-point response to your comments, including the specific changes made, is provided below:

Major changes:

1. Given the NIDs in China are required to be reported within 24 hours and the Reports are published monthly, it would be better to include a longer duration (e.g. 6 more months) of incidence data of those 24 NIDs after March 2023 in the analyses. This would allow them to check whether the seasonality patterns changed after the epidemic, particularly if wanted to claim the statement in the last paragraph of Discussion section.

Author response:

We greatly appreciate your suggestion and concur that a more extensive duration of incidence data would indeed provide valuable insights into potential shifts in seasonality patterns post-epidemic. With this in mind, we have expanded our study period to encompass January 2008 through December 2023.

We have organized this timeframe into several distinct segments corresponding to critical points in the epidemic timeline: Firstly, the period from January 2008 to December 2019 serves as the pre-epidemic period, offering a snapshot of the epidemiological landscape prior to the advent of SARS-CoV-2. The onset of the epidemic and the subsequent response through PHSMs are partitioned into several phases. The initial phase, PHSMs I, extends from January 2020 to March 2020, reflecting the early response to the outbreak. This phase was succeeded by PHSMs II, from April 2020 to October 2022, characterized by sustained measures aimed at controlling viral spread. The demarcation between PHSMs I and PHSMs II primarily hinges on the cessation of lockdowns in Wuhan, signaling a significant shift in control strategies. The subsequent period, from November 2022 to January 2023, corresponds to the Chinese government's transition away from the "Dynamic COVID-zero" strategy, marking the beginning of the epidemic period. The final phase, extending from February 2023 to December 2023, is acknowledged as the post-epidemic period. This period is marked by a significant decrease in the positive rate of COVID-19 tests, indicating a reduction in viral transmission.

In the second part of our results section, we analyzed the differences in the adjusted IRR between the PHSMs and epidemic periods, as compared to the post-epidemic period (in line 228-244). We found all IIDs experienced adjusted IRR lower than 1, expected hepatitis E, which initially had a median adjusted IRR of 0.91 (IQR: 0.77-0.99, $P < 0.001$) during PHSMs and epidemic periods, but in the post-epidemic period, the median adjusted IRR of rose to 1.17 (IQR: 1.04-1.28, $P = 0.05$), indicating a rebound in cases (Fig. 6A). Additionally, HFMD (Fig. 5A), AHC (Fig. 5A), and hepatitis B (Fig. 5B) showed signs of resurgence. AHC witnessed a substantial outbreak in September 2023, with reported cases reaching 125,264, a count surpassed only by September 2010 (Supplementary Appendix 3). This outbreak predominantly affected the southern provinces, with Guangdong being the epicenter. (Supplementary Fig. 4 in Supplementary Appendix 1). Additionally, Pertussis also raised alarms, registering the highest incidence since 2008, with 9,126 cases from October to December 2023, concentrated in Guangdong province (Supplementary Fig. 17 in Supplementary Appendix 1, Supplementary Appendix 3).

We have also supplemented our findings with an examination of changing trends in the post-epidemic period (in line 210-227). This comprehensive approach ensures a thorough understanding of the evolving epidemiological landscape throughout the COVID-19 pandemic and beyond.

2. Authors used first 10 years (2008-2017) data as training set and 2 more years (2018-2019) data as a testing set to define the model for forecasting the rate during PHSMs and endemic period. More information should be provided on the definition of optimal model and how reliable of comparing the results fitted by the different model for different NIDs?

Author response:

Thank you for raising this point. We apologize for not providing a detailed explanation of the optimal model selection and the reliability of comparing results for different NIDs. In our study, we evaluated multiple models for forecasting NIDs such as the neural network model, Bayesian structural time series model, prophet model, exponential smoothing (ETS) model, seasonal autoregressive integrated moving average (SARIMA) model, and hybrid model that combine SARIMA, ETS, STL (seasonal and trend decomposition using loess), and neural network components to the training dataset. The model performance was evaluated based on a comprehensive standardized index, which was calculated based on RMSE, MAPE, and SMAPE. Additionally, these indicators, each with unique range sensitivities, underwent a transformation process to standardize their values for comparative analysis using z-normalization. We selected the model with the greater comprehensive standardized index as the optimal model for forecasting the incidence after January 2020 and included a description of these metrics in the manuscript to better explain our model selection process in the method section (line 594-618).

To ensure the reliability of our comparative analysis across different models for various NIDs, we implemented cross-validation techniques of our evaluation process (line 616-619 in method section). This method involved partitioning the training data into multiple subsets to facilitate both model fitting and subsequent evaluation of these distinct subsets, as detailed in Supplementary Table 4 of Supplementary Appendix 3. The training datasets ranged from 7 to 10 years, with the test datasets being established 2 years after the training dataset. To mitigate the impact of the 2019 rubella outbreak on the test dataset, the rubella data from 2019 were excluded (line 204-209 in the result section). Our findings reveal that the optimal model identified for each disease consistently exhibited a superior comprehensive standardized index, irrespective of variations in the size of the training set (Supplementary Table 4 in Supplementary Appendix 2). This approach not only emphasizes the reliability of our model selection process but also enhances the credibility of our results by demonstrating the stability and accuracy of the chosen models across different datasets. This rigorous validation method provides a solid foundation for our analysis, ensuring that our comparative results are both reliable and reflective of the underlying data characteristics.

Some results were quite surprising or unexpected. Using rubella as an example, due to the vaccination, rubella has reached at very low incidence rate since 2013 but suddenly had a high pick in 2019 then back to nearly zero level since 2020 (Supplementary Figure 24). The paper didn't explain the unusual peak in 2019 and the forecasting model repeated that

unusual pattern during 2020-2023 (Figure 4) by the optimal model selected (Bayesian structural model as shown in Figure 3).

Author response:

We apologize for the oversight. The surge in rubella cases in 2019 can be ascribed to various factors. One plausible reason is the decline in MMR and MR vaccination coverage that year, leading to an upsurge in susceptible individuals and subsequent outbreaks. We have detailed this in the results section (lines 326-332). To mitigate the impact of the unexpected rubella outbreak on the model construction, we excluded the data from 2019 in the first-stage model testing set, revealing ETS as the top-performing model (Fig. 3P). The ETS model was trained in the second stage using data from 2008-2018, and despite the exclusion of 2019 data, the rubella predictions persistently showed seasonal peaks ranging from several hundred to one thousand cases (Fig. 4P).

Considering ETS, SARIMA, and the hybrid model, which exhibited strong performance on the 2018 test dataset with close composite standardized indices (1.92 vs. 1.73 vs. 1.63) (Fig. 3R), we proceeded to forecast rubella incidence from 2019 to 2023 using these models. The outcomes consistently indicate a marked decrease in rubella prevalence, as indicated by the median of the adjusted IRR dropping below 0.2 during the public health emergency preparedness and response periods and the epidemic phase (Supplementary Fig. 49 in Supplementary Appendix 1). We have integrated this information into the discussion section (lines 283-290).

3. Different scales used for the Y-axis in Figure 4 for different disease which may exaggerate the effects for those low prevalent NIDs.

Author response:

We appreciate your review and highlighting the concern related to the use of different scales for the Y-axis in Figure 4. We acknowledge that this can potentially result in an exaggeration of effects for low prevalent NIDs. We apologize for this oversight in the figure presentation. In the revised version of the paper, we have added Figure 5 and carefully addressed this issue by using consistent scales for the Y-axis across 24 NIDs. This will ensure that the effects are accurately represented without any exaggeration. Thank you for bringing this to our attention.

Specific comments:

1. Page 4 line 65: please indicate when, where and what the data source are?

Author response:

Thank you for your comment. We apologize for the lack of information regarding the data source in the last paragraph of the background section. In line 57-63, we have revised the paper to include the relevant details, including specifying the time, location, and source of the data used in our study.

2. Page 4 line 79: ‘The last reported’ should be ‘The least reported’?

Author response:

Thank you for pointing out this mistake. Yes, you are correct. The phrase 'The last reported' should indeed be changed to 'The least reported'. We apologize for the error and will make the necessary corrections in the revised version of the paper in line 72.

3. Page 4 line 80 and other relevant places in the context: please keep consistent for using the term, either as ‘zoonotic infectious disease’ or ‘natural focal disease’

Author response:

Thank you for your comment. We apologize for the inconsistency in the use of these terms. We will review the paper and make sure to use zoonotic infectious disease and ZIDs consistently throughout the text, including this one and other relevant places.

4. Page 5 and other relevant places in context: the specific supplementary figure need to be referred clearly in the text

Author response:

Thank you for your comment. We acknowledge that there were instances where we failed to refer clearly to specific supplementary figures in the text. We apologize for this oversight and have revised the manuscript to ensure that all supplementary figures are appropriately referenced throughout the text. All supplementary figures and supplementary tables referred to specific numbers in the main content.

5. I found it's hard to follow the results presented in different orders of NIDs categories in the context as well as figures, better to keep in a consistent order

Author response:

Thank you for your feedback. We apologize for any confusion caused by the inconsistent order of NIDs categories in the results and figures. We agree that maintaining a consistent order would enhance the readability of the paper. In response to your suggestion, we have revised the paper to ensure that the NIDs categories are presented in the same order throughout. This includes reorganizing the figures and rephrasing the corresponding text to align with the consistent order.

6. Page 7 in the section of 'respiratory infectious diseases': there was an abnormal peak of rubella in 2019 which should be presented/explained (Supplementary Figure 24)

Author response:

We thank the reviewer for pointing out the abnormal peak of rubella in 2019. We apologize for not including the explanation and supplementary figure in the initial submission. In the revised version of the paper, we have added Supplementary Fig. 16 and Supplementary Fig. 40 in supplementary appendix 1, which displayed the temporal and spatial variation between 2008 and 2023 in the Chinese mainland. We have included an explanation of the abnormal peak in the result section (line 117-125) and discussion section (line 326-332), respectively.

7. Page 7 in the section of 'respiratory infectious diseases': what about tuberculosis which showed a regular pattern as in Supplementary Figure 22

Author response:

Thank you for bringing up this point. We apologize for not including tuberculosis in the discussion of respiratory infectious diseases on page 7. Tuberculosis is indeed an important respiratory infectious disease, and its regular pattern can be observed in Supplementary Fig. 13 and Supplementary Fig. 39 in supplementary appendix 1. We have included tuberculosis and its corresponding pattern as shown in supplementary figures and updated the result section (line 123-126) and discussion section (line 419-433), respectively.

8. Pages 7-8 lines 143 onwards: confused, not sure whether PHSMs period I and II were counted as 'before' or 'during' epidemic as data from both periods been presented in both sections?

Author response:

Thank you for your comment and concern. We apologize for any confusion caused. To clarify and shorten the title to meet the requirements of Nature Communication, we have modified the title of the article to "**Temporal shifts in 24 notifiable infectious diseases in China before and during the COVID-19 pandemic**", where "before the COVID-19 pandemic" refers to before December 2019, also known as the pre-epidemic period in this article, and "during the COVID-19 pandemic"

refers to the period from January 2020 to December 2023. The period during the COVID-19 pandemic has been divided into the following time frames: the PHSMs period I (from January 2020 to March 2020), the PHSMs period II (from April 2020 to October 2022), the epidemic period (November 2022 to January 2023), and the post-epidemic period (February 2023 to December 2023). These details have been added in the method section (lines 519-538). 519-538

Originally intended to summarize this chapter and bridge the transition, we have moved this paragraph to the beginning of "Short-term trends of NIDs during the COVID-19 pandemic" to avoid any misunderstandings (line 152).

9. Page 8 second paragraph: not sure why present certain NIDs optimal model but not others? And seems dysentery has two models been selected as optimal model: neural network (shown in both text and Figure 3) or hybrid model (shown in text line 162)

Author response:

We appreciate the reviewer's comment. The challenge in determining the most suitable model for predicting dysentery could have arisen from the calculation method of the composite standardized index. This issue was pinpointed during a thorough manual examination, leading us to refine the calculation approach of the composite standardized index by incorporating root mean square error (RMSE), mean absolute percentage error (MAPE), and symmetric mean absolute percentage error (SMAPE). As a result of these adjustments, the ETS model is now identified as the optimal choice for forecasting dysentery cases (refer to Fig. 3C).

10. Page 10 lines 201-205: malaria was missed

Author response:

Thank you for pointing out this oversight. We apologize for the error in our manuscript. In the revised version, we have corrected the information regarding malaria and have added additional clarification in the result section (line 263-264).

11. Page 10 first paragraph: clarify what's the information shown in Figure 5 A C E G?

Author response:

We apologize for the lack of clarity in the description of Figure 5 A C E G in the first paragraph on page 10. Figure 5 has been moved to Figure 6 which displays the distribution of adjusted IRR during different periods. Panel A, C, E, and G represent the adjusted IRR of IIDs, BSTDs, RIDs, and ZIDs, respectively. We have updated the figure legend to provide a more detailed explanation of the information shown in Figure 5 A C E G and described these panels in the results section (line 228-244).

12. Page 11 line 227: Fig. 5 I should be Fig. 5 J?

Author response:

Thank you for bringing this to our attention. We apologize for the mistake in the figure labeling. We have updated the figure and carefully checked all figure labeling in our paper.

13. Page 11 line 232: not sure about the statement of HFMD 'significant susceptibility to the impact of PHSMs' which perhaps in absolute term as a highly prevalent NID, but not in relative term, due to a similar pattern found as in the pre-epidemic period

Author response:

We appreciate the reviewer's comment. Our statement regarding HFMD's susceptibility to the cluster analysis and distribution of adjusted IRR. To clarify, we have revised our statement to "HFMD was the only disease with a high incidence and susceptibility to PHSMs" in the result section (lines 256-258) and discussion section (lines 368).

14. Page 13 lines 264-267: surprised that with such low incident rate, malaria showed a surprisingly high relative correlation (Figure 6). Moreover, other three NIDs had only very weak correlation (just above 0.2) as shown in Figure 6, thus the statement sounds not very reliable

Author response:

We appreciate the reviewer's comment and agree that the implications of the weak correlations for the other NIDs should be discussed. In the revised version, we have adjusted the cross-correlation analysis to examine the relationship between PHSMs stringency index and the incidence reduction of identified NIDs. We found that malaria showed a moderate correlation coefficient of 0.42 without lag time, and HFMD, mumps, malaria, and JE exhibited either a moderate or weak correlation without lag time. However, dengue fever, rubella, scarlet fever, and pertussis showed no association with PHSMs stringency index with the absolute number of correlation coefficient below 0.2.

We have revised the description in the discussion section (line 283-286) and provided a more in-depth analysis of HFMD (line 303-312), mumps (line 335-339), malaria (line 311-314) and JE (line 325-329).

15. Page 13 lines 275-277: there's an abnormal peak in 2019 which authors didn't present however the forecast repeated the wave of that, thus the statement may not reliable

Author response:

We apologize for not including the abnormal peak of rubella in 2019 in the presentation. This peak was indeed observed, and we have included the abnormal peak in the result section (line 204-209) of the paper. This rubella outbreak occurred from March to June 2019, with a nationwide cumulative report of 25,736 cases (Fig. 2F, Supplementary Fig. 40). This outbreak led to an increase in reported cases in all provinces except for Zhejiang, Tibet, Tianjin, and Qinghai (Supplementary Fig. 16), with most cases concentrated in Chongqing, Hunan, Guangdong, Sichuan, and Gansu, where Chongqing and Hunan experienced the most severe outbreaks, accumulating 4,334 and 3,733 reported cases within four months respectively (Fig. 2F, Supplementary Table 2).

To avoid the impact of the 2019 rubella outbreak on model validation and re-training, we excluded the data from 2019 in the testing set of the first phase and the training set of the second phase. However, even with this exclusion, the difference between observed data and forecasted outcomes revealed a significant impact on rubella during this period (Fig. 4P). This content is provided in the result section (line 204-209) and Supplementary Fig. 50 in supplementary appendix 1.

16. Page 16 line 344: not sure how author defined 'high-incidence NIDs' in the study, some NIDs had very low rate

Author response:

We apologize for the confusion. In our study, 24 'high-incidence NIDs' refer to infectious diseases that have a higher incidence compared to other NIDs not included in the study. We defined this based on cumulative cases of more than 20,000 between 2008 to 2023 (Supplementary Appendix 3). For further elucidation, we have included the "disease selection criteria" in the methodology section to explicitly describe the specific criteria used for disease selection.

17. Pages 16-17 lines 345-350: not sure about this discussion point, and more information needs to be provided for the NIDs reporting system in China

Author response:

Thank you for raising this concern about the third limitation in the discussion section. What we wanted to express is that time series models are not suitable for every notifiable infectious disease. Although we selected the optimal model based on the composite standardized index for disease analysis, we are not sure if the choice of model is related to the type of disease. For example, intestinal infectious diseases can be analyzed using ETS or SARIMA models. Of course, considering this is minor and the limitation of article length, we have removed this point in the revised manuscript.

We also appreciate the reviewer's comment and acknowledge the significance of furnishing additional details concerning the National Notifiable Diseases Surveillance System (NNDSS), commonly known as the NIDs reporting system in China. In our revised manuscript, we have expanded the method section discussing China's NNDSS (line 490-508), providing a more detailed description of its coverage, history, and relevant characteristics. This will help readers gain a better understanding of the context in which our study is conducted and the implications of our findings in relation to the NIDs reporting system.

18. Page 17 361 onwards: don't think this study support this argument, if wanted, the author should include those two diseases in the analyses and compare the data before and after epidemic

Author response:

Thank you for your comment. We appreciate your suggestion to include the analysis of those two diseases and compare the data before and after the pandemic. We agree that this would provide a more comprehensive understanding of the impact of the epidemic. However, as *Mycoplasma pneumoniae* is not currently classified as a Notifiable Infectious Disease (NID), and because influenza, whose detection depends on specialized sentinel surveillance systems, was not included in this study, we are unable to supplement data on these two diseases. By supplementing NIDs data from August 2023 to December 2023, we found that hand, foot, and mouth disease (HFMD), acute hemorrhagic conjunctivitis (AHC), and pertussis all experienced a resurgence during the post-epidemic period (February 2023 to December 2023). Additionally, literature has shown that population susceptibility to influenza increased by 140.1% and 74.8% in southern and northern China, respectively. We believe that this information would help support our argument.

1. Wang Q, et al. Increased population susceptibility to seasonal influenza during the COVID-19 pandemic in China and the United States. *J Med Virol* 95, e29186 (2023).

19. Page 19 lines 398-402: how comparable of those two datasets? More information should be provided

Author response:

We appreciate the reviewer's comment regarding the comparability of the two datasets. In the revised version of the paper, we have added panel A in Supplementary Fig. 1-24 in supplementary appendix 1 specifically discussing the comparability of the datasets. We have provided detailed information on the similarities and differences between the datasets in the data collection of method section (line 490-508). This additional information should address the reviewer's concern.

The monthly NIDs reports aggregate data from the NNDSS based on the reported date and are published by the National Health Commission of China, which updates very quickly, usually updating the previous month's report the following month. The data provided by the Chinese Public Health Science Data Center (https://www.phsciencedata.cn/share/ky_sjml.jsp) also aggregates data from the NNDSS based on the onset date but is updated at a slower pace, taking around 3-4 years from the onset date of cases. For HFMD, AHC, infectious diarrhea, mumps, rubella, echinococcosis, and typhus from January 2008 to February 2009, the monthly NIDs report was not included, and we used data provided by the Chinese Public Health Science Data Center instead. After comparing the incidence from the two data sources from 2010 to 2021, no significant differences were found (Supplementary Fig. 1, S2, S4, S14, S16, S22, S23).

20. Page 19 line 411: better to include data up to October 2023 in the analyses

Author response:

Thank you for your suggestion. We agree that including data up to December 2023 would enhance the comprehensiveness of our analyses. We have reviewed our data collection process and have found that we already have access to the data for that period. Therefore, we will update our analyses to include data up to December 2023 in the revised version of the paper.

21. Page 20 second paragraph: reference 15 is an editorial, not relevant

Author response:

We apologize for the error in referencing. Upon reevaluating the content, we agree that reference 15 is not relevant to the discussion. We will remove this reference from the paper.

22. Page 23 line 482-484: more explanations are needed for the formula included; please also clarify how the optimal model has been defined/selected?

Author response:

We acknowledge the reviewer's comment and apologize for the lack of clarity in explaining the formula included. In the revised version of the paper, we will provide a detailed explanation of the formula, including the variables used and their significance in the context of our study (line 621-627).

In our paper, the optimal model was defined and selected based on the composite standardized index, including accuracy, efficiency, and interpretability. We employed the greater composite standardized index, that integrated root mean square error (RMSE), mean absolute percentage error (MAPE), and symmetric mean absolute percentage error (SMAPE). The details of this process have been elaborated in the revised version of the paper to provide a clear understanding of how the optimal model was defined and selected in the method section (line 594-619).

Reviewer #3 (Remarks to the Author):

In this paper, the authors conduct a modeling study using different types of time-series models to model transmission dynamics of 24 notifiable infectious diseases (NIDs), categorized into 4 groups according to the mode of transmission that were reported in mainland China from January 2008 to March 2023 – before and after the COVID-19 pandemic measures (PHSMs). They describe disease trends, identify the best forecasting model, and evaluate the impacts of the COVID-19 PHSMs on each of the 24 NIDs. Authors conclude that while PHSMs can be an effective short-term solution for inadvertently controlling the spread of non-SARS-CoV-2 NIDs, they are not effective or sustainable in the long-term, and the development and implementation of effective vaccines should be prioritized.

This is a comprehensive study on NID trends with different modes of transmission before and during the COVID-19 pandemic and the implementation of PHSMs in China. The methods chosen are described in detail, clearly, and extensively. Results and figures are well represented. The result section shows the number of disease cases, long-term trends before the COVID-19 pandemic, and describes seasonality for each of the 24 NIDs. Six different time-series models were implemented, and the best model was identified for each disease. These models were then used to forecast the incidence of NIDs assuming the absence of SARS-CoV-2 in 2020 and onward and compare predicted incidence with the real incidence data. Models were used to describe diseases most affected by PHSMs. Discussion should be developed further to make this paper much more impactful.

Author response:

We are grateful for your constructive feedback on our manuscript. Your comments have been essential in identifying aspects of our study that could be enhanced, especially in developing the discussion section to amplify the impact of our findings. In line with your advice, we have revised the manuscript to enrich the discussion on the implications of our results, including:

1. Elucidating the changes that occurred during the PHSMs II period in the discussion section.
2. Expanding the discussion to include references to publications and news from other countries.
3. Engaging professional manuscript polishing services to improve the language and clarity.

A detailed point-by-point response to your comments, including the specific changes made, is provided below:

Major changes:

Lines 287-298: There is a substantial decrease in the incidence during the first half of PHSM I, but then the trends start going back towards the baseline trends during PSHM II (except for respiratory viruses), which is not specifically discussed or clearly explained in this paragraph. Omicron BA.2 appeared in Dec 2021, much later than April 2020, which was the start of the PSHM II period, to explain this observation.

Author response:

We apologize for the confusion. In our study, January 2020 to March 2020 marks PHSMs I, reflecting the early response to the outbreak. This is followed by the period from April 2020 to October 2022, termed PHSMs II, characterized by sustained measures aimed at controlling the spread of the virus. The distinction between PHSMs I and PHSMs II is primarily based on the cessation of lockdowns in Wuhan, marking a significant shift in control strategies. In this paragraph, we aim to highlight the substantial decline observed during both PHSMs period I and the epidemic period. To prevent any misunderstandings, we have reorganized the paragraph structure and specified the respective time frames (lines 518-532).

-
1. Geng MJ, et al. Changes in notifiable infectious disease incidence in China during the COVID-19 pandemic. *Nat Commun* 12, 6923 (2021).

How about discussing the stringency index here? Perhaps, although still in place, measures were more relaxed during PSHM II and contributed to this. The paragraph has no supporting references. Comparing to studies from other countries could be also beneficial.

Author response:

Thank you for your constructive suggestion. We have taken it into account and expanded our discussion on the stringency index in the pertinent segment of our manuscript (line 533-542). This index serves as a valuable tool in assessing the severity of measures implemented during the PSHM II period. To provide a comprehensive understanding, we've explained that the stringency index is a composite measure that is based on several indicators such as school closures, workplace closures, and travel bans, among others. Each indicator is given a value between 0-100 (100 being the strictest). The stringency index thus offers a quantifiable measure to compare how strictly different countries or regions have responded to the pandemic over time. Our literature review has further identified several implications associated with the relaxation of PSHMs. We found that easing these measures has led to increased social interactions and population migration, subsequently enhancing the potential for infectious disease transmission. This phenomenon has been documented not only in China but also in other countries such as the UK and New Zealand. These findings underscore the delicate balance governments must maintain between mitigating the spread of the virus and minimizing socio-economic disruptions.

1. Huang QS, et al. Impact of the COVID-19 nonpharmaceutical interventions on influenza and other respiratory viral infections in New Zealand. *Nat Commun* 12, 1001 (2021).
2. Santana C, Botta F, Barbosa H, Privitera F, Menezes R, Di Clemente R. COVID-19 is linked to changes in the time-space dimension of human mobility. *Nat Hum Behav* 7, 1729-1739.
3. Huang QS, et al. Impact of the COVID-19 nonpharmaceutical interventions on influenza and other respiratory viral infections in New Zealand. *Nat Commun* 12, 1001 (2021).

Lines 324-338: Proposed explanations for the observed patterns in this paragraph lack references that can provide support. There should be some available in the recent literature.

Author response:

Thank you for your feedback. We acknowledge the need for references to support the proposed explanations in the paragraph. We will update the paragraph with relevant references from the recent literature to strengthen the support for the explanations provided (line 385-391).

For dengue fever and malaria, there is currently more literature supporting the effective control of international epidemics, which has reduced the incidence of the diseases. This is mainly because the transmission of these two diseases relies on mosquitoes, and in most areas of China, the climate is predominantly subtropical and temperate, preventing mosquitoes from overwintering on the Chinese mainland, thus limiting local transmission. Furthermore, China implements quarantine policies for incoming individuals during Public Health and Social Measures periods, ranging from 10-14 days, while the incubation periods for dengue fever and malaria are mostly 8-12 days and 7-30 days, respectively. This means that most cases of malaria and dengue fever can be detected during the quarantine of incoming individuals, further suppressing local transmission in China.

For brucellosis, we have consulted experts in CDC and veterinary medicine. Apart from inadequate animal vaccination and a meat shortage that spurred the expansion of sheep herds and, practitioners, there has been an increase in home poultry farming. The literature supporting this is extremely limited, but news reports during the impact of COVID-19 have shown a rise in backyard chickens, which can support our perspective.

-
1. Geng, Meng-Jie, et al. "Changes in notifiable infectious disease incidence in China during the COVID-19 pandemic." *Nat. Commun.*, vol. 12, no. 6923, 26 Nov. 2021, pp. 1-11, doi:10.1038/s41467-021-27292-7.
 2. WHO. "Dengue and severe dengue." *World Health Organization: WHO*, 17 Mar. 2023, www.who.int/news-room/fact-sheets/detail/dengue-and-severe-dengue.
 3. CDC. "CDC - Malaria - About Malaria - Disease." 8 Feb. 2009, www.cdc.gov/malaria/about/disease.html.
 4. Yang, Huimin, et al. "Epidemic Characteristics, High-Risk Areas and Space-Time Clusters of Human Brucellosis — China, 2020–2021." *CCDCW*, vol. 5, no. 1, 6 Jan. 2023, pp. 17-22, doi:10.46234/ccdcw2023.004.
 5. Chappell, Bill. "'We Are Swamped': Coronavirus Propels Interest In Raising Backyard Chickens For Eggs." *NPR*, 3 Apr. 2020, www.npr.org/2020/04/03/826925180/we-are-swamped-coronavirus-propels-interest-in-raising-backyard-chickens-for-egg.

Minor changes:

Lines 52, 55, and 60 and some other lines throughout the manuscript would benefit from making some grammar changes.

Author response:

Thank you for pointing out the grammar issues in the specified lines and throughout the manuscript. We apologize for any confusion caused by these errors. We have revised the highlighted sections and sought language polishing services to enhance the manuscript's clarity and readability.

Line 64 – add “time-series” to clarify the type of models in the introduction.

Author response:

Thank you for your suggestion. We have revised this in the introduction to include the term "time series" to clarify the type of models being discussed. The revised line now reads as follows: " using time series models to predict transmission trends without PHSMs or epidemics." (line22-23).

Lines 129-131: The statement is confusing; I am guessing that you mean the inflection point in 2018 was mainly due to the upsurge of bloodborne and STD cases in 2018, while the other 3 disease groups maintained their fluctuating patterns and disease incidence levels.

Author response:

We apologize for any confusion caused by our statement. Your understanding is correct. After analyzing the incidence data, we found the inflection point shown in 2017, which primarily attributed to the significant increase in BSTDs, whereas the other three disease groups exhibited fluctuating patterns and maintained their previous levels of disease incidence. We have revised the statement and described the temporal trend of 5 BSTDs in the result section (line 111-116).

Line 211 – Would it not now shift from “relaxed intercity travel restrictions” since it has shifted from “dynamic zero-COVID” policy to relaxed intercity travel restrictions in April 2020 (described on line 185)?

Author response:

Thank you for bringing up this point. We apologize for the error. In April 2020, China began implementing the "dynamic zero-COVID" policy. Therefore, it should be corrected to say "with the adoption of the 'dynamic zero-COVID' policy and the relaxation of intercity travel restrictions by the Chinese government" in the earlier part of the sentence (line 187-189). This marks the beginning of PHSMs period II. In the latter part of the sentence, the Chinese government shifted from the

"dynamic zero-COVID" policy to adopting a new approach aimed at coexisting with the virus during the epidemic period (line 210-212). We have revised the manuscript to accurately reflect this policy shift.

Line 688 – Is this a typographical error? Dysentery, not dynthesis

Author response:

Thank you for catching that typo. You are correct, it should be dysentery and not dynthesis. We apologize for the error and will make the necessary corrections in the final version of the paper.

Figure 1A – spelling of “cumulative”

Author response:

Thank you for pointing out the spelling error. We apologize for the mistake. We have corrected the spelling of "cumulative" and replaced the circle graph with a tree map in Figure 1A.

REVIEWERS' COMMENTS

Reviewer #1 (Remarks to the Author):

I appreciate the authors' diligent efforts in addressing my comments. The introduction on methodology has been enhanced, exhibiting greater clarity. The findings appear more sensible. I have no further comments on this paper.

Reviewer #2 (Remarks to the Author):

There are big improvements of the manuscript in this revision, including extended the post-pandemic duration in the analyses and further detailed and clear description of the methods and other information. Some minor comments to consider:

1. Line 171: It's not consistent with Fig 3 in which the Hybrid method marked as 'Optimal Model for (I) Syphilis too';
2. Line 212-214: Can't follow the description of this sentence when comparing to Figure 6
3. Lines 214-216: Not clear, please revise the sentence
4. Line 273: it was '5 periods' in the revision, please also cross-check and correct other parts in the context, e.g. line 623 and Figure Legends part
5. Pages 30-31 - 'Statistical analyses': please clarify whether 'incidence' refers to 'number of new cases' or the 'incidence rate'? Also please indicate the adjustment factors implemented when calculating 'Adjusted IRR'?
6. Line 661: a typo of 'thought'?
7. Line 897: Figure 7 B-C are not cluster trees?
8. Line 884: Figure 6 (Legend) is up to 'Dec 2023', not to 'July 2023'

Reviewer #3 (Remarks to the Author):

The present study is relevant as it demonstrates the impact of Public Health and Social Measures (PHSMs) on 24 Notifiable Infectious Diseases (NIDs) in China. The findings reveal that these measures had the greatest effect on reducing the incidence of respiratory NIDs, while their impact on bloodborne and sexually transmitted diseases was more moderate. Dengue fever and malaria were also affected, primarily due to travel restrictions. These results align with studies conducted in other countries. The methodology is robust, with sufficient detail provided for reproducing the study.

The revised manuscript has effectively addressed my previous comments and clarified points that needed further explanation. The discussion is now well-developed, integrating analyses from other countries to establish connections between studies and compare outcomes. Overall, the manuscript has significantly improved since the previous version, and additional factors relevant to the observed incidence trends have been incorporated into the discussion. Therefore, I recommend accepting the manuscript for publication.

Reviewer #3 (Remarks on code availability):

Reviewed the script coded in R (not the one in Python). The code includes a README file with clear instructions. I was able to install and run the code.

REVIEWERS' COMMENTS

Reviewer #2 (Remarks to the Author):

There are big improvements of the manuscript in this revision, including expanded the post-pandemic duration in the analyses and further detailed and clear description of the methods and other information. Some minor comments to consider:

1. Line 171: It's not consistent with Fig 3 in which the Hybrid method marked as 'Optimal Model for (I) Syphilis too;

Author response:

Thank you for pointing out the inconsistency between the description on line 171 and Figure 3 regarding the Hybrid method's designation as the "Optimal Model for (I) Syphilis." We apologize for this oversight. Thank you for pointing out the problem, we've made the changes in the article, the optimal model for Syphilis we did look at wrong when analyzing Figure 3, so we have adjusted it in the article Your attention to detail is greatly appreciated and will contribute to the accuracy of our manuscript.

2. Line 212-214: Can't follow the description of this sentence when comparing to Figure 6

Author response:

We apologize for the confusion caused by the description in line 212-214. We have carefully reviewed the sentence and the corresponding Figure 6. Brucellosis saw its adjusted IRR drop to 0.58 in December 2022 should be Figure 6H instead of Figure 6G, and we have fixed this mistake in line 212-213.

3. Lines 214-216: Not clear, please revise the sentence

Author response:

Thank you for your comments on lines 214-216. We will be revising this sentence to improve its clarity and readability. Thank you for your guidance, which will help us improve the manuscript.

4. Line 273: it was '5 periods' in the revision, please also cross-check and correct other parts in the context, e.g. line 623 and Figure Legends part

Author response:

Thank you for identifying the mention of '5 periods' in the revision. We apologize for the error and have made the necessary correction. The revised version now accurately reflects the correct number of periods.

5. Pages 30-31 – 'Statistical analyses': please clarify whether 'incidence' refers to 'number of new cases' or the 'incidence rate'? Also please indicate the adjustment factors implemented when

calculating 'Adjusted IRR'?

Author response:

Thank you for your question. It is here that "incidence" does not refer to 'number of new cases' or the 'incidence rate', it's just 'incidence'. When calculating the 'Adjusted IRR,' no adjustment factors were utilized; instead, we applied Laplace smoothing to adjust the IRR calculations.

6. Line 661: a typo of 'thought'?

Author response:

The word "thought" was indeed a typo, and we've corrected it, thanks for your review!

7. Line 897: Figure 7 B-C are not cluster trees?

Author response:

No, Figure 7 B-C are both cluster scatterplots. Instead of cluster trees, Figure C is a partial enlargement of Figure B.

8. Line 884: Figure 6 (Legend) is up to 'Dec 2023', not to 'July 2023'

Author response:

Yes, it should be up to December 2023 here, we've made the correction in the main text, thanks for the careful review!